# Pre-existing cross-reactive immunity to highly pathogenic avian influenza 2.3.4.4b A(H5N1) virus in the United States

Zhu-Nan Li[1], Feng Liu [1], Yu-Jin Jung[1], Stacie Jefferson[1], Crystal Holiday[1], F. Liaini Gross[1], Wen-Pin Tzeng[1], Paul Carney[1], Ashley Kates [2], Ian A. York [1], Nasia Safdar[2], James Zhou[3], Marie-jo Medina[3], Vittoria Cioce[3], Christine M. Oshansky[3], C. Todd Davis[1], James Stevens[1], Terrence Tumpey[1] & Min Z. Levine [1] ✉

The unprecedented 2.3.4.4b. A(H5N1) outbreak in dairy cattle, poultry, and spillover to humans in the United States (US) poses a major public health threat. Population immunity is a critical component of influenza pandemic risk assessment. We assessed the pre-existing cross-reactive immunity to 2.3.4.4b A(H5N1) viruses and analyzed 1794 sera from 723 people (0.5–88 yrs) in multiple US geographic regions during 2021–2024. Pre-existing neutralizing and hemagglutinin (HA)-head-binding antibodies to A(H5N1) were low, but there were substantial cross-reactive binding antibodies to N1 neuraminidase (NA) of 2.3.4.4b A(H5N1). Antibodies to group 1 HA stalk were also prevalent and increased with age. A(H1N1)pdm09 infection and influenza vaccination did not induce neutralizing antibodies to A(H5N1) viruses but induced significant rise of functional NA inhibition (NAI) antibodies to N1 of 2.3.4.4b A(H5N1), and group 1 HA stalk antibodies. Moreover, pre-pandemic stockpiled 2.3.4.4c vaccine can elicit cross-reactive neutralizing antibodies to 2.3.4.4b A(H5N1) viruses. Understanding population susceptibility is essential for pandemic preparedness.

Since their emergence almost 3 decades ago[1], highly pathogenic avian influenza (HPAI) A(H5) viruses have become endemic in wild birds and poultry in many countries, causing sporadic human infections, most from exposure to infected poultry[2]. In recent years, spillover infections of HPAI A(H5) viruses from birds and poultry to mammals have occurred in increasing numbers of mammalian species with potential mammal-to-mammal transmission[3]. Clade 2.3.4.4b HPAI A(H5N1) viruses were first detected in wild birds in the United States (US) in 2021, and since have caused widespread outbreaks in US poultry[4]. The first A(H5N1) human case in the US was identified in 2022 from direct exposure to 2.3.4.4b H5N1 infected poultry[5].

Since March 2024, an unprecedented outbreak has been ongoing among US dairy cattle in multiple states caused by 2.3.4.4b HPAI A(H5N1) viruses[6]. These viruses have transmitted among cattle, from cattle back to poultry, and spilled over into humans[6–8]. In March 2024, A(H5N1) human infection was identified in a Texas dairy farm worker after exposure to presumably infected dairy cattle, a HPAI 2.3.4.4b H5N1 virus was isolated, marking the first known case of cattle-to-human transmission of an avian influenza A virus[6,9]. As of September 10, 2025, 70 A(H5N1) human cases have been confirmed in multiple US states from the current outbreak. Most (65/70) were associated with exposure to infected dairy cattle or poultry, and all were caused by clade 2.3.4.4b HPAI A(H5N1) viruses[10], raising concerns of the pandemic risk of these viruses[11].

Influenza pandemics can occur when novel influenza viruses that can infect humans mutate to gain sustained transmissibility among

[1]Influenza Division, Centers for Disease Control and Prevention, Atlanta, GA, USA. [2]Department of Medicine, University of Wisconsin, Madison, WI, USA. [3]Biomedical Advanced Research and Development Authority, Washington, DC, USA. ✉e-mail: mlevine@cdc.gov

populations that have little or no pre-existing immunity. Pre-existing host immunity is a critical parameter in assessing susceptibility to novel avian and other emerging influenza virus infection and the risk of an influenza pandemic[11]. Most of the US population has pre-existing immunity to seasonal influenza viruses from exposure through infection and vaccination with these viruses. However, it is largely unknown whether there were antibodies to the emerging 2.3.4.4b viruses in the population. Furthermore, unlike the A(H5) cases previously reported from outbreaks outside the US, which often caused severe disease with a high rate of mortality (50%)[12], thus far, most reported A(H5N1) human cases from the current outbreak in the US have had mild clinical symptoms. Several recent studies suggested that pre-existing immunity could impact responses to 2.3.4.4b A(H5N1) infection[13–15], whether pre-existing host immunity plays a role in disease severity needs to be better understood. Multiple-immune mechanisms could contribute to protection against influenza virus infection[16]; most of the past studies of human immunity to A(H5) focused on the antibodies targeting the hemagglutinin (HA) head of the virus. Little is known about the population immunity to other immunological targets of the A(H5) viruses that could also contribute to protection and attenuation of disease, including antibodies to neuraminidase (NA) and the HA stalk.

In this study, we assessed the levels of neutralizing antibodies, HA head binding antibodies, functional NA inhibition antibodies (NAI), NA binding antibodies to HPAI clade 2.3.4.4b A(H5N1) viruses, and antibodies to group 1 HA stalk using 1794 sera collected from 723 participants (aged 0.5–88 yrs) from multiple geographic regions in the US across recent influenza seasons (2021–2024). We investigated whether there were pre-existing cross-reactive immunity to 2.3.4.4b A(H5N1) viruses in the US population, and whether current seasonal influenza A virus infection or seasonal influenza vaccination can induce antibodies that are cross-reactive to 2.3.4.4b A(H5N1) viruses. Lastly, we also assessed the ability of the pre-pandemic stockpiled 2.3.4.4c A(H5) vaccine to elicit cross-reactive antibody responses to these newer 2.3.4.4b viruses.

## Results

### Low prevalence of antibodies to the HA head of clade 2.3.4.4b A(H5N1) viruses in the US

We have been monitoring the population immunity in the US to both seasonal and novel influenza viruses in a multi-season longitudinal study. From November 2021 to February 2023, 489 participants from all 10 US Department of Health and Human Services (HHS) regions (Fig. 1a and Table 1) and 9 age groups (1–4 yrs, 5–10 yrs, 11–20 yrs, 21–30 yrs, 31–40 yrs, 41–50 yrs, 51–60 yrs, 61–70 yrs and > 70 yrs, Fig. 1b) were enrolled and provided sera at 3 waves (time points) across the 2021-23 influenza seasons (Table 1 and Fig. 1c–e). A total of 1467 sera were analyzed by a 12-plex multiplex influenza antibody detection assay (MIADA) (Supplementary Table S1). High levels of binding antibodies to the HA head of a representative seasonal A(H1N1)pdm09 virus A/Wisconsin/588/2019 were detected across most age ranges (Fig. 1c–e). In contrast, there were very low levels of pre-existing binding antibodies to the HA head of the HPAI A(H5N1) 2.3.4.4b virus A/American wigeon/South Carolina/22-000345-001/ 2021 (AW/2021) during 2021–23 influenza seasons (p < 0.004, Fig. 1c–e).

As a part of pandemic preparedness, we also evaluated sera collected from a different population of healthy adults who received 2 doses of either MF59 or AS03 adjuvanted A/gyrfalcon/WA/410886/ 2014/H5N8 (GF/2014) (clade 2.3.4.4c) vaccine from National Pre-Pandemic Influenza Vaccine Stockpile (NPIVS) by microneutralization (MN) assays. Vaccination with stockpiled older 2.3.4.4c GF/2014 A(H5) vaccine induced robust cross-reactive neutralizing antibodies to newer 2.3.4.4b A(H5N1) AW/2021 virus (Fig. 2a), with 57% (42/74) of post-H5 vaccination sera had cross-reactive neutralizing antibody titers ≥40 to the wild type 2.3.4.3b A(H5N1) AW/2021 virus (Fig. 2a, b). A(H5) 2.3.4.4c vaccination also induced highly correlated binding antibodies to the HA head of the AW/2021 and A/Texas/37/2024 (the virus isolated from the first cattle-to-human spillover human case in the current outbreak) 2.3.4.4b viruses (r = 0.9791, Fig. 2c). The neutralizing antibody titers

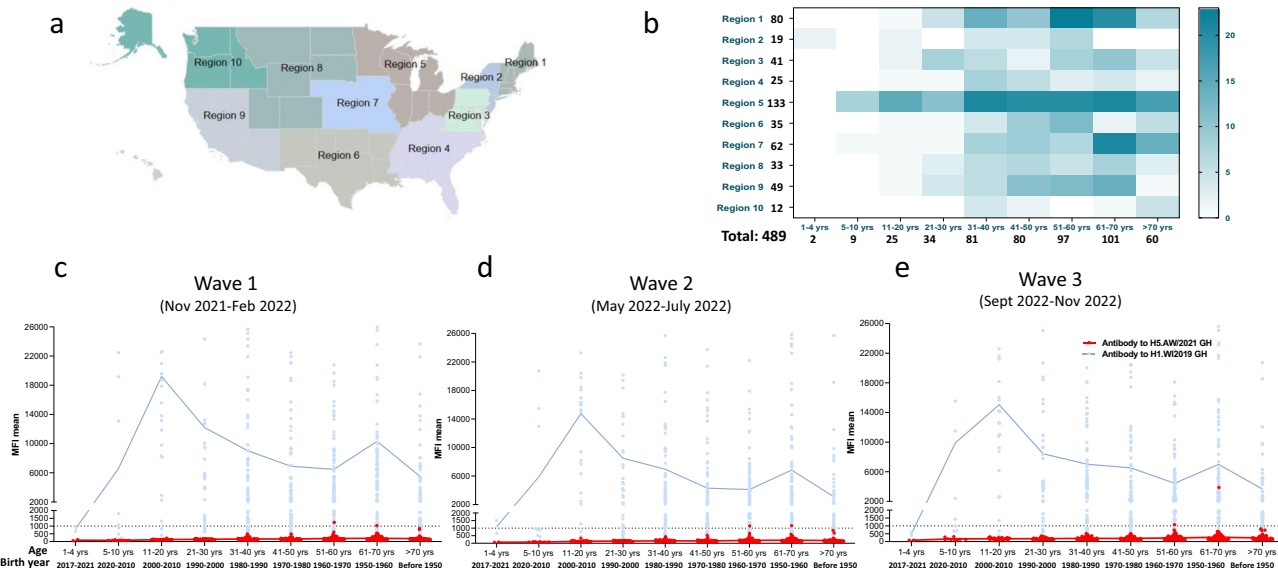

**Fig. 1 | Age-stratified prevalence of binding antibodies to the hemagglutinin (HA) head of 2.3.4.4b HPAI A(H5N1) and A(H1N1)pdm09 viruses in the United States in 2021–23. a** Ten HHS regions in the US. **b** Distribution of the regions and ages of 489 participants enrolled in the longitudinal population immunity study, 3 waves (time points) of sera were collected from each participant from 2021 to 2023 influenza seasons. **c–e** Scatter plots of age-stratified prevalence of binding antibodies to the HA head of 2.3.4.4b A(H5N1) A/American wigeon/South Carolina/22-000345-001/2021 (AW/2021) and A/Wisconsin/588/2019 H1N1pdm09 (Wis/2019)

by waves (total 1467 sera collected from 489 individuals from 3 waves). Connected lines represent the mean antibody MFI of each age group, dotted lines represent MFI of 1000. **c** Wave 1 (Nov 2021–Feb 2022) n = 489 participants. **d** Wave 2 (May 2022–July 2022) n = 489 participants; (**e**) Wave 3 (Sept 2022–Nov 2023) n = 489 participants. Wilcoxon matched-pairs signed rank test was performed to compare the MFI values between Wis/2019 H1 and AW/2021 H5 HA head per age group in each wave. P < 0.004 for all the age groups with n ≥ 3 (except the 1–4 years group n = 2, sample size too small for statistical analysis).

**Table 1 | Characteristics of the participants and sera evaluated in the study**

| Sera | | Number of participants | Region/States | Timing of the collection | Age range (median) yrs | S1 (days post influenza vaccination - range (median) | S2 (days post influenza vaccination or infection - range (median) | Paired sera | Serum number | Serum number by season | Serum number by study type |
|---|---|---|---|---|---|---|---|---|---|---|---|
| US population immunity longitudinal (2021-23 seasons) | Wave 1 (2021-2022 season) | 489 | HHS region 1-10 | Nov 2021-Feb 2022 | 1-88 (51) yrs | N/A | N/A | N | 489 | 1467 sera from 489 participants (2021-23 seasons) | 1794 sera from 723 participants |
| | Wave 2 (2021-22 season) | | HHS region 1-10 | May 2022-July 2022 | 1-88 (51) yrs | N/A | N/A | N | 489 | | |
| | Wave 3 (2022-23 season) | | HHS region 1-10 | Sept 2022-Nov 2022 | 1-88 (51) yrs | N/A | N/A | N | 489 | | |
| Seasonal Influenza (2023-24 season) | Vaccination (pre & post vaccination) Pediatrics (0.5-3 yrs) IIV4 recipients 2023-24 | 19 | TX, CA, LA | 2023-2024 | 0.5-2.8 (1.0) yrs | 0 | 21-21 (21) | Pre/Post | 38 | 327 sera from 234 participants (2023-24 season) | |
| | Adult (18-49 yrs) IIV4 recipients 2023-24 | 20 | TX, CA, LA | 2023-2024 | 20-48 (36) yrs | 0 | 21-24 (21) | Pre/post | 40 | | |
| | Older adult (≥65 yrs) IIV4 recipients 2023-24 | 20 | TX, CA, LA | 2023-2024 | 65-92 (70) yrs | 0 | 21-26 (22) | Pre/post | 40 | | |
| | Infection (RT-PCR confirmed influenza positive) A(H1N1) infection | 21 | AZ, MO, OH, PA, TX | Oct 2023-Jan 2024 | 20-36 (46) yrs | 0-7 (3) | 21-49 (26) | Acute/convalescent | 42 | | |
| | | 41 | AZ, MI, MO, OH, TX | Oct 2023-Feb 2024 | 19-74 (41) yrs | 0-7 (3) | N/A | Acute only | 41 | | |
| | A(H3N2) infection | 13 | AZ, MO, OH, PA | Oct 2023-Jan 2024 | 18-75 (27) yrs | 0-7 (2) | 19-45 (24) | Acute/convalescent | 26 | | |
| | | 22 | AZ, MI, MO, OH, PA | Dec 2023-Feb 2024 | 18-77(23) yrs | 0-7 (2) | N/A | Acute only | 22 | | |
| | Controls (influenza negative) Controls (RT-PCR confirmed influenza negative) | 78 | MI, OH, PA | Oct 2023-Feb 2024 | 18-77 (37) yrs | yrs | N/A | Acute only | 78 | | |
| A(H5) 2.3.4.4c HPAI | Vaccination Gry/WA 2.3.4.4b A(H5) vaccine with MF59 or ASO3 | 74 | N/A | 2018-19 | 18-49 yrs | N/A | 43 (22 days post 2nd dose) | N | 74 | 74 | 74 |
| **Total number of participants** | | **797** | | | | | | | **Total number of sera** | | **1868** |

TX Texas, CA California, LA Louisiana, AZ Arizona, MO Missouri, OH Ohio, PA Pennsylvania, MI Michigan.

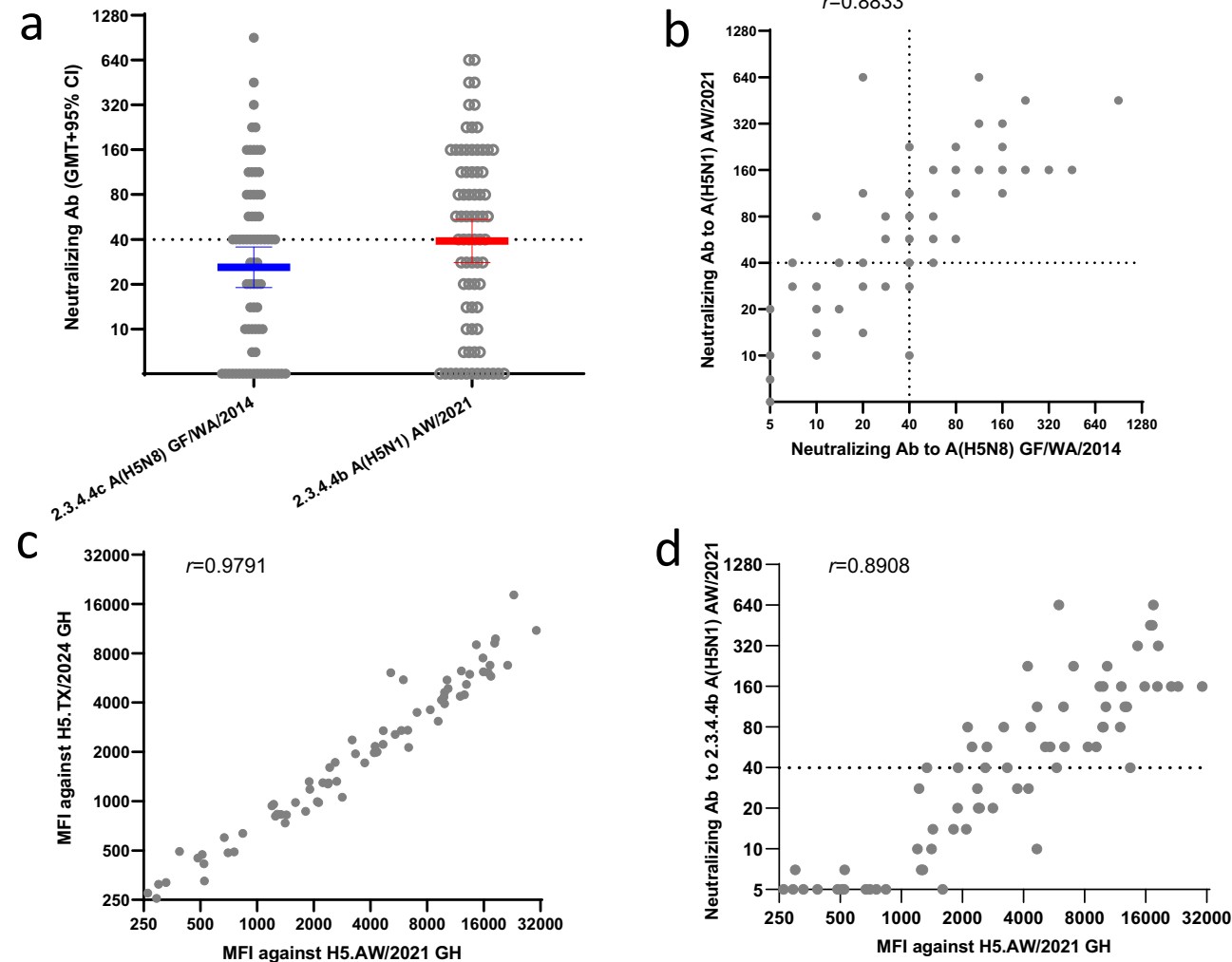

**Fig. 2 | Vaccination with MF59 or AS03 adjuvanted A/gyrfalcon/Washington/410886/2014 H5N8 vaccine induced cross-neutralizing and binding antibody responses to newer 2.3.4.4b HPAI A(H5N1) viruses. a** Total 74 post-A(H5N8) vaccination sera were tested by microneutralization (MN) assays, Scatter plot of neutralizing antibody titers against wild type A/gyrfalcon/Washington/410886/2014 (H5N8) (GF/WA/2014) and A/American wigeon/2021 (H5N1) (AW/2021), geometric mean titers (GMTs) were shown with error bars indicate 90% confidence interval (CI). Two-tailed Wilcoxon matched-pairs signed rank test was performed to compare the MN titers between GF/WA/2014 and AW/2021 viruses, $p = 0.0003$. Spearman's rank correlation analysis was performed for (**b**–**d**). **b** Correlation of the neutralizing antibodies to A(H5N1) AW/2021 vs A(H5N8) GF/WA/2014; (**c**) Total 74 post- A(H5N8) vaccination sera were tested by 28-plex MIADA assay, the correlation of binding antibodies to globular head (GH) HA of 2.3.4.4b A(H5N1) AW/2021 and A/ Texas/37/2024; **d** The correlation between neutralizing antibody titers and binding antibodies (MFI) to the HA head of 2.3.4.4b A(H5N1) AW/2021.

against wild type 2.3.4.4b A(H5N1) AW/2021 virus correlated with binding antibodies to the HA head of the same virus (Spearman's rank correlation $r = 0.8908$): higher neutralizing antibody titers correlated well with higher median fluorescence intensities (MFIs), however some samples with high MFI (binding antibody) did not show functional neutralizing antibody titers (MN ≤ 10) (Fig. 2d).

Using the MFIs of the 42 post-A(H5) vaccination sera with MN titer ≥40 to the wild-type 2.3.4.4b A(H5N1) AW/2021 virus as positive sera and 1467 sera from the 2021–23 population immunity study collected before the current A(H5) outbreak as negative sera (Table 1), we analyzed the sensitivity and specificity of the MIADA assay in detecting HA head binding antibodies to 2.3.4.4b A(H5N1) AW/2021. At various MFI cutoffs from 500–2000 to 2.3.4.4b A(H5N1) AW/2021 GH, the assay had good sensitivity and specificity, with the highest J-index (sensitivity+specificity-1) at MFI cutoff of 1000 (Supplementary Table S2).

Based on the World Health Organization (WHO) A(H5) case definition[17], seropositivity to A(H5) virus using a single serum sample was defined as a neutralizing antibody titer ≥40 to an A(H5) virus that is

antigenically similar to the viruses that are causing the current outbreak, and either hemagglutination inhibition (HI) titer to an antigenically similar A(H5) virus ≥40, or a positive result from an A(H5)-specific immunological assay such as a multiplex antibody binding assay. Seropositive sera that have antibodies specific to A(H5N1) viruses indicate serological evidence of infection. To investigate whether there is any seropositivity, 12 samples from the 2021-23 population immunity study (Fig. 1) that had the highest MFI values to 2.3.4.4b A(H5N1) AW/2021 GH were tested by both MN and HI assays against wild type 2.3.4.4b HPAI A(H5N1) AW/2021 virus and 2.3.4.4b HPAI A(H5N1) A/Texas/37/2024 virus. All sera were negative (<10) against both wild type A(H5N1) viruses in both assays, although most had pre-existing neutralizing and HI antibodies to circulating seasonal A(H1N1) pdm09 viruses (Table 2). These results confirmed no A(H5N1) seropositivity (i.e., seronegative) among 1467 US population immunity sera surveyed across 3 time points during 2021-23 seasons (Fig. 1), with no serological evidence of A(H5N1) infection.

Since the onset of the A(H5N1) outbreak in dairy cattle in early 2024, we updated the MIADA assay to a high throughput 28-plex with

**Table 2 | Microneutralization assays and hemagglutination inhibition assays against wild type 2.3.4.4b HPAI H5N1 viruses to confirm no A(H5) seropositivity of 2021-23 sera with high MFIs to 2.3.4.4b H5N1 viruses**

| Sera | 2.3.4.4b A(H5N1) | | | | | | Seasonal A(H1N1)pdm09 (control) | | |
|---|---|---|---|---|---|---|---|---|---|
| | HA head of A/American wigeon/SC/22-00345-001/2021 H5N1 | A/American wigeon/SC/22-00345-001/2021 H5N1 wild type virus | | A/Texas/37/2024 wild type H5N1 virus | | | HA head of A/Wisconsin/588/2019 A(H1N1)pdm09 virus[4] | A/Victoria/2570/2019 A(H1N1)pdm09 virus[4] | |
| | MFI in MIADA (Binding antibody) | Neutralizing Antibody[3] | HI antibody[3] | Neutralizing antibody[3] | HI antibody[3] | | MFI in MIADA (Binding antibody) | Neutralizing Antibody[3] | HI antibody[3] |
| 1 | 3889 | 5 | 5 | 5 | 5 | | 9680 | 226 | 160 |
| 2 | 1239 | 5 | 5 | 5 | 7 | | 2649 | 20 | 10 |
| 3 | 1159 | 5 | 5 | 5 | 5 | | 12,724 | 320 | 80 |
| 4 | 1139 | 5 | 5 | 5 | 7 | | 2337 | 20 | 7 |
| 5 | 1072 | 5 | 5 | 5 | 7 | | 2149 | 20 | 5 |
| 6 | 1025 | 5 | 5 | 5 | 5 | | 26,084 | 1280 | 320 |
| 7 | 852 | 5 | 5 | 5 | 7 | | 1532 | 20 | 10 |
| 8 | 842 | 5 | 5 | 5 | 7 | | 3031 | 113 | 40 |
| 9 | 807 | 5 | 5 | 5 | 7 | | 1298 | 20 | 10 |
| 10 | 785 | 5 | 5 | 5 | 5 | | 2393 | 14 | 10 |
| 11 | 716 | 5 | 5 | 5 | 5 | | 1870 | 5 | 5 |
| 12 | 678 | 5 | 5 | 5 | 7 | | 1724 | 7 | 5 |
| A(H5N1) Positive control[1] | 13,988 | 5120 | 5120 | 2560 | 5120 | | 36 | 5 | 5 |
| Negative control[2] | 30 | 5 | 5 | 5 | 5 | | 26 | 5 | 5 |

[1]Positive control serum: Serum from ferrets infected with A/American wigeon/SC/22-00345-001/2021 2.3.4.4b H5N1 virus.
[2]Negative control serum: Serum from influenza-negative naïve ferrets.
[3]Sera were tested at 1:10 predilution in MN and HI assays, titers less than 10 are reported as 5. Multiple replicates were performed for each sample in MN and HI assays against each virus, geometric mean titer (GMT) from multiple replicates is reported as the final titer for each sample in MN and HI assays.
[4]A/Wisconsin/588/2019 and A/Victoria/2570/2019 are antigenically-like A(H1N1)pdm09 viruses representing the circulating seasonal A(H1N1)pdm09 viruses during the study season.

an expanded antigen cocktail including HA and NA of the virus isolated from 2.3.4.4b HPAI A(H5N1) A/Texas/37/2024 for antibody landscape analysis (antigens shown in Supplementary Table S1), and assessed recent sera collected in the 2023-24 influenza season (327 sera collected from 234 participants aged 0.5-77 yrs) (Table 1). Similar to data from previous seasons, all 2023-24 season sera were low for total binding antibody to 2.3.4.4b A(H5N1) AW/2021 GH (MFI < 1000, Fig. 3b), MFIs to A(H5N1) A/Texas/37/2024 GH were also low (<1000, Fig. 3a) across all age groups, suggesting the cross-reactive immunity to the HA head of the 2.3.4.4b viruses remained low in the 2023-24 season. As expected, binding antibodies to the representative seasonal A(H1N1)pdm09 virus A/Wisconsin/67/2022 GH were high across most age groups (Fig. 3a, b).

**Seasonal influenza virus infection and vaccination did not induce neutralizing antibodies to 2.3.4.4b virus or binding antibodies to 2.3.4.4b A(H5N1) virus HA head**

Next, we assessed whether seasonal influenza A virus infection or influenza vaccination could induce cross-reactive antibodies to 2.3.4.4b A(H5N1) viruses. We collected acute (within 7 days from symptom onset) and convalescent (19-49 days since symptom onset) sera from reverse transcription polymerase chain reaction (RT-PCR) confirmed influenza A(H1N1)pdm09 and A(H3N2) virus infections in the 2023-24 season and analyzed both neutralizing and binding antibody responses to seasonal and A(H5N1) viruses. Seasonal influenza A virus infection induced a significant rise of neutralizing antibodies to the infecting A(H1N1)pdm09 (Fig. 4a) or A (H3N2) (Fig. 4b) viruses, but no neutralizing antibodies (MN < 10) to AW/2021 2.3.4.4b A(H5) virus were detected in any of the acute and convalescent sera (Fig. 4a, b). We also analyzed pre- and post-seasonal influenza vaccine sera collected from older adults

(≥65 yrs), adults (18–49 yrs), and pediatric (0.5–3 yrs) populations who received 2023–24 inactivated quadrivalent influenza vaccines (IIV4). IIV4 seasonal influenza vaccination induced a significant rise in neutralizing antibodies to A(H1N1)pdm09 vaccine virus, but no neutralizing antibodies (MN < 10) to 2.3.4.4b A(H5N1) AW/2021 virus were detected in all pre- and post-vaccination sera from all age groups (Fig. 4c–e).

In the HA binding antibody landscape analysis by the MIADA assay, seasonal A(H1N1)pdm09 or A(H3N2) virus infection or IIV4 vaccination in 2023–24 not only induced significant (p < 0.05) rise of binding antibodies to the HA head of the seasonal virus of exposure (either the infecting virus or antigens in the quadrivalent vaccines), in most cases it also back-boosted antibodies to older seasonal viruses within the same subtype (Fig. 5). None, however, induced a significant rise in cross-reactive binding antibodies to the HA head of –either AW/2021 or A/Texas/37/2024 2.3.4.4b HPAI A(H5N1) viruses (Fig. 5a–e).

**Pre-existing functional NAI antibodies and binding antibodies to the N1 neuraminidase of 2.3.4.4b A(H5N1) virus in the US population from past exposure to seasonal A(H1N1)pdm09 viruses**

We analyzed the prevalence of cross-reactive antibodies to the N1 neuraminidases of two 2.3.4.4b viruses in sera collected from 2023 to 2024 participants (327 sera from 234 participants). Surprisingly, there were substantial levels of pre-existing binding antibodies to the N1 neuraminidases of both 2.3.4.4b A(H5N1) viruses (AW/2021 and A/Texas/37/2024) across most age groups >18 yrs. The youngest children (0.5-3 yrs) had the lowest and older adults (>70 yrs) had the highest pre-existing binding antibodies to the N1 neuraminidase of 2.3.4.4b A(H5N1) (Fig. 3c, d).

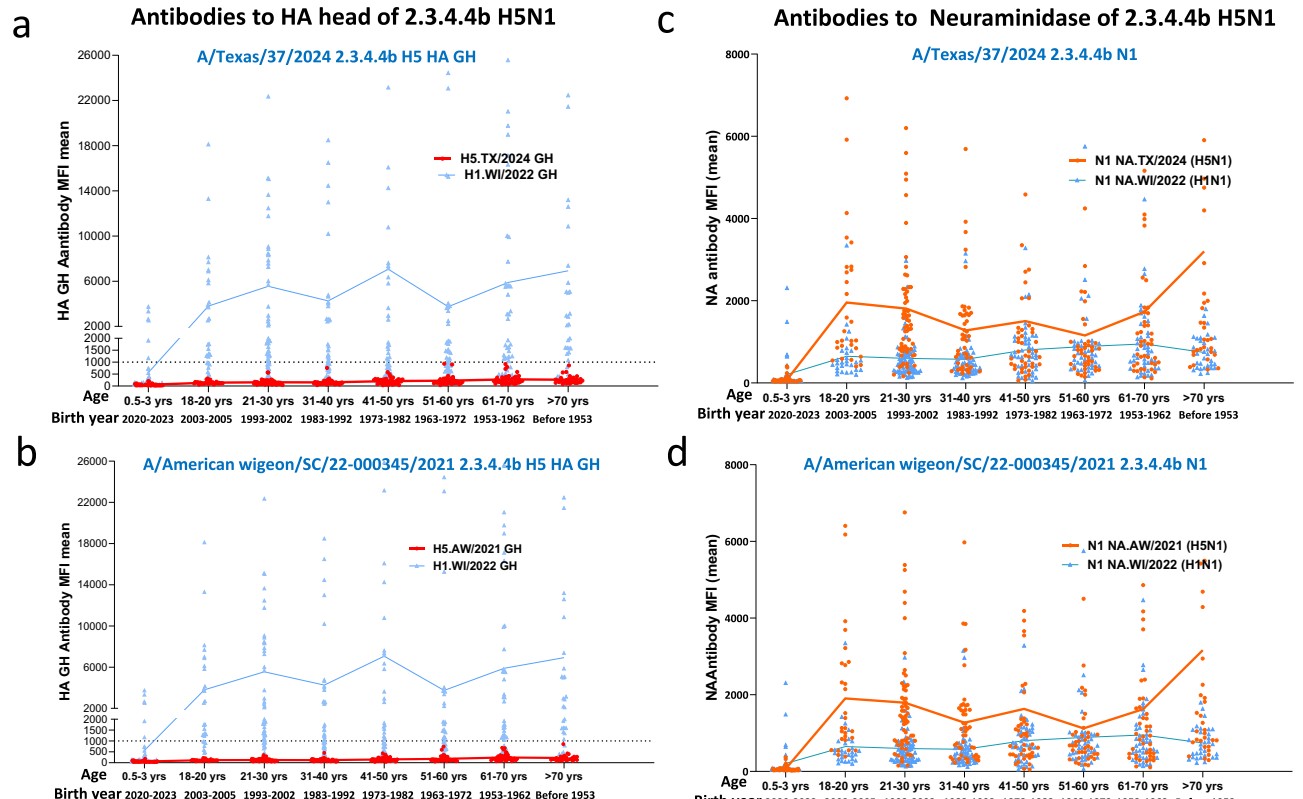

**Fig. 3 | Age-stratified prevalence of binding antibodies to the hemagglutinin (HA) head, neuraminidase of HPAI 2.3.4.4b A(H5N1) and A(H1N1)pdm09 viruses in the US population in 2023-24.** Total 327 sera collected from 234 participants during 2023-24 influenza season were stratified into 8 age groups. Age-stratified scatter plot of binding antibodies: (**a**) Antibodies to HA globular head (GH) of 2.3.4.4b A(H5N1) A/Texas/37/2024 and A(H1N1) pdm09 A/Wisconsin/67/2022 (WI/

2022) (**b**) Antibodies to HA GH of 2.3.4.4b A/American wigeon/SC/22-000345/2021 (AW/2021) and A(H1N1)pdm09 WI/2022; (**c**) Antibodies to N1 neuraminidase of 2.3.4.4b A(H5N1) A/Texas/37/2024 and A(H1N1) pdm09 WI/2022; (**d**) Antibodies to N1 neuraminidase of 2.3.4.4b AW/2021 and A(H1N1)pdm09 WI/2022. Connected lines represent mean antibody MFI of each age group, dotted line in (a) and (b) represent MFI of 1000.

We then investigated whether the high levels of pre-existing antibodies to N1 of A(H5N1) were due to cross-reactivity from seasonal influenza virus A virus infection or vaccination. NA binding antibody landscape analysis using the acute and convalescent sera collected from seasonal influenza infections showed that A(H1N1)pdm09 virus infection in the 2023-24 season not only boosted N1 binding antibodies to recent circulating seasonal A(H1N1)pdm09 viruses, it also induced significant rises in N1 binding antibodies to both 2.3.4.4b A(H5N1) viruses (AW/2021 and A/Texas/37/2024) ($p < 0.05$) (Fig. 6a), whereas A(H3N2) virus infection did not boost significant rises in N1 binding antibodies to A(H5N1) ($p > 0.05$) (Fig. 6b). Following seasonal IIV4 vaccination in 2023-24 in 3 age groups (older adult ≥65 yrs, adult 18–49 yrs, and pediatric 0.5-3 yrs), there were some rises of N1 binding antibodies to 2.3.4.4b A(H5N1) viruses in adult and older adult groups, but these were not statistically significant (Fig. 6c–d).

Next, to assess the functional NAI antibodies, we constructed H6Nx reverse genetic viruses with mismatched H6 (to eliminate potential interference from HA antibodies in the human sera) and NA from seasonal A(H1N1)pdm09 (A/Wisconsin/588/2019), A(H3N2) (A/Darwin/6/2021), and A(H5N1) (N1 from HPAI 2.3.4.4b A/Texas/37/2024) viruses to assess the NAI antibody responses from seasonal influenza exposure by Enzyme-linked lectin assay (ELLA). 2023–24 seasonal A(H1N1)pdm09 virus (with N1 NA) infection, but not A(H3N2) (with N2 NA) infection, induced a significant rise of functional cross-reactive NAI antibodies targeting N1 of 2.3.4.4b A/Texas/37/2024 (H5N1) virus (Fig. 7a, b). Furthermore, 2023–24 seasonal influenza IIV4 vaccination also induced significant NAI antibody responses to N1 of 2.3.4.4b A/Texas/37/2024 A(H5N1) virus in older adult (≥ 65 yrs), adult (18-49 yrs), and pediatric (0.5–3 yrs) participants (Fig. 7c–e).

## Prevalence of antibodies to the group 1 HA stalk in the US population and the impact of seasonal influenza vaccination and infection on antibodies to the group 1 HA stalk

Influenza viruses are classified into two groups based on their antigenic properties of HA: group 1 and group 2. Both A(H5) and A(H1) belong to group 1 HA. HA stalks from A(H5N1) and A(H1N1)pdm09 are highly conserved (Supplementary Fig. 1) thus antibodies targeting the HA-stalk are cross-reactive[18].

We assessed the prevalence of antibodies to the group 1 HA stalk (headless H1 stalk from A/Michigan/15/2015 H1N1). There was a wide range of pre-existing anti group 1 HA stalk antibodies in the population (Fig. 8a). In sera collected in 2023-24, when stratified by age groups with 2–10-year age intervals, the mean MFI to group 1 HA stalk antibody in each age group increased with age, with the lowest prevalence of group1 HA stalk antibody in very young children (0.5–3 yrs) and highest in older adults >70 yrs, reflecting the accumulation of past exposures to group 1 influenza viruses (Fig. 8a).

Lastly, to investigate the source of the anti HA stalk antibodies, we analyzed antibody responses to the group 1 HA stalk following seasonal influenza vaccination and infection. In the 3 age groups that received 2023–24 IIV4 vaccination, pre-existing group 1 HA stalk antibody levels increased with age, young children (0.5–3 yrs) had the lowest and older adults (≥ 65 yrs) had the highest antibodies to the group 1 HA stalk. IIV4 vaccination elicited significant rise of group 1 HA stalk antibodies in adults (18–49 yrs) ($p < 0.05$) (Fig. 8b). Furthermore, adults infected with A(H1N1)pdm09 (group 1 virus) ($p < 0.05$), but not those infected with A(H3N2) (group 2 virus) ($p > 0.05$), mounted significant rise of group 1 HA stalk antibody responses (Fig. 8c). Of note, A(H1N1)pdm09 infection induced higher rise of group 1 HA stalk antibodies (mean fold rise in

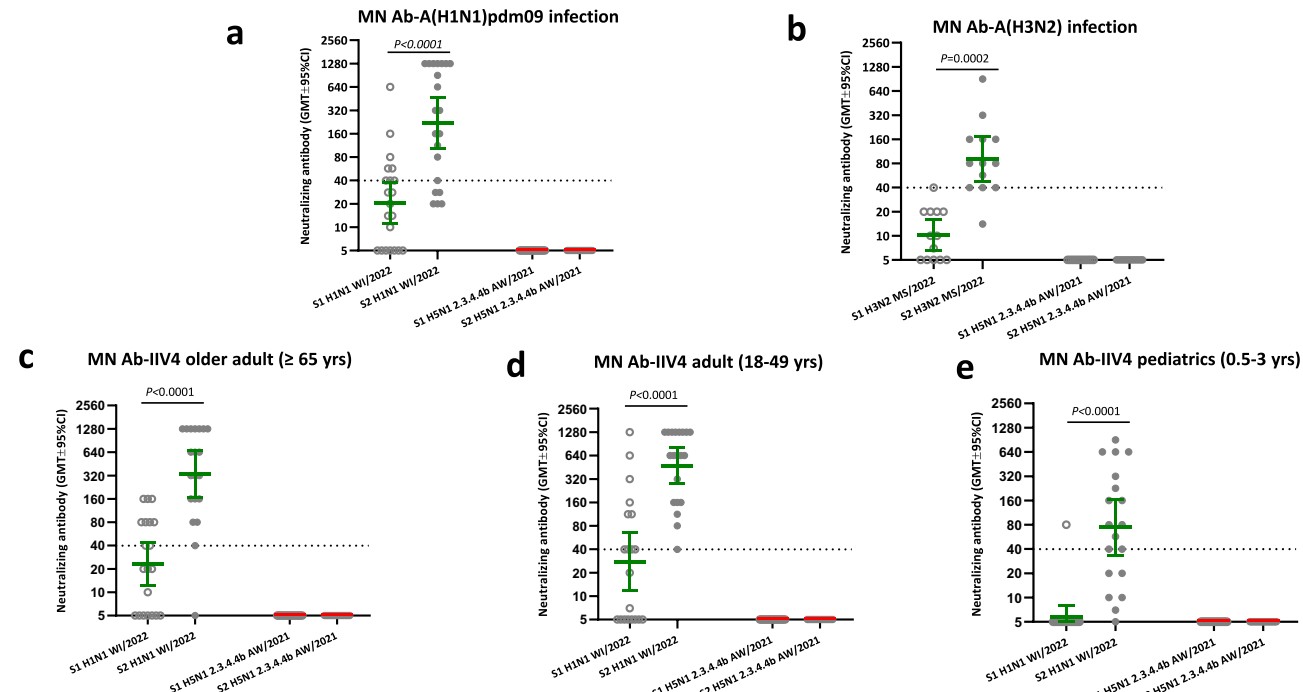

**Fig. 4 | Seasonal influenza infection and vaccination in 2023–24 did not induce neutralizing antibodies to 2.3.4.4b HPAI A(H5N1) virus.** Paired serum samples collected from seasonal influenza infection and quadrivalent inactivated influenza vaccination (IIV4) participants during 2023–24 influenza season were tested by microneutralization (MN) assays. Neutralizing antibody titers against 2.3.4.4b A(H5N1) A/American wigeon/SC/22-000345/2021 (AW/2021) candidate vaccine virus (CVV) and seasonal influenza A(H1N1) pdm09 A/Wisconsin/67/2022 (WI/2022) were plotted in scatter plots. **a** Acute (S1) and convalescent (S2) sera from A(H1N1) pdm09 infection (*n* = 21 participants); **b** Acute (S1) and convalescent (S2) sera from A(H3N2) infection (*n* = 13 participants); **c** Pre (S1) and post (S2) vaccination sera collected from older adults (≥ 65 yrs) who received IIV4 (*n* = 20 participants); (**d**) Pre (S1) and post (S2) vaccination sera collected from adults (18-49 yrs) who received IIV4 (*n* = 20 participants); (**e**) Pre (S1) and post (S2) vaccination sera collected from pediatric participants (0.5–3 yrs) who received IIV4 (*n* = 19 participants). Y axis: geometric mean titers (GMTs) with error bars indicate 95% confidence interval (CI) of neutralizing antibodies. Dashed line: Neutralizing antibody titer of 40 threshold. Statistical analyses were conducted using two-tailed Wilcoxon matched-pairs signed rank test, *p* < 0.05 is considered statistically significant.

convalescent/acute sera: 2.8) than IIV4 vaccination (mean fold rise in post/ pre-vaccination sera: 1.9) (Fig. 8b, c) in adults.

## Discussion

Population immunity against new emerging novel viruses is a key factor for influenza pandemic risk assessment[19]. Amid the ongoing A(H5N1) outbreaks in cattle and poultry and the continued spillover to humans, our study provides a timely assessment of the cross-reactive immunity present in the US population to 2.3.4.4b A(H5N1) viruses. Results from the current study demonstrate that the levels of the pre-existing neutralizing antibodies and the HA head binding antibodies to 2.3.4.4b A(H5) viruses in the US population are low, consistent with previous reports of low seroprevalence (mostly measured by HI antibodies) even in populations at increased risk of A(H5) exposure (e.g., poultry workers)[12]. However, our study revealed that there were substantial levels of pre-existing antibodies to the N1 of 2.3.4.4b A(H5N1) virus, and group 1 HA stalk antibodies in an age-related pattern in the US population. Furthermore, these pre-existing cross-reactive immunities to A(H5) virus (group 1) were most likely from past exposures to seasonal A(H1N1)pdm09 (group 1), not A(H3N2) (group 2) viruses.

While neutralizing antibodies targeting the HAs of the influenza virus are the main correlate of protection in reducing the risk of influenza virus infections, multiple immune mechanisms can contribute to protection from influenza[16]. Although seasonal influenza A(H1N1)pdm09 virus infection and influenza vaccination did not induce neutralizing and HA head binding antibodies to A(H5N1) viruses (Figs. 4, 5), both could induce a significant increase of cross-reactive functional NAI antibodies to the N1 neuraminidase of 2.3.4.4b A(H5N1) (Fig. 7). A(H1N1)pdm09 infection also induced significant increase of

the cross-reactive N1 binding antibodies to 2.3.4.4b A(H5N1) (Fig. 6). Following seasonal influenza vaccination, although there is also an increase of the mean MFI of the cross-reactive N1 binding antibodies to A(H5N1), it is not statistically significant, in part, due to the wide range (confidence intervals) of the N1 binding antibodies (Fig. 6). Sequence analysis showed that there is significant genetic distance between the HA head of the 2.3.4.4b A(H5N1) and A(H1N1)pdm09 viruses, with amino acid homology at approximately only 53%, and differences across multiple antigenic sites (Supplementary Table S3 and Supplementary Fig. 2). In contrast, there is a higher level of amino acid sequence homology (86–88%) between the N1 sequences of 2.3.4.4b A(H5N1) and recent circulating seasonal A(H1N1)pdm09 viruses (Supplementary Table S4).

Neuraminidase antibodies have been considered as an independent correlate to protect against influenza[20]. Although they cannot prevent infection, neuraminidase antibodies can prevent virus egress, reduce viral shedding, and thus could attenuate disease and lessen disease severity[21–23]. Past studies have demonstrated that seasonal influenza virus infection and vaccination could induce cross-reactive and protective NA-specific antibodies, conferring prophylactic protection against older avian A(H5N1) viruses in mice and ferret models[24–26]. There were 10 amino acid differences between the N1 of 2.3.4.4b A/Texas/37/2024 H5N1 and A/Wisconsin/67/2022 (H1N1) pdm09 viruses among the putative NA antigenic sites (Supplementary data Figs. 3 and 4). A previous epitope mapping study showed that three primary amino acids on N1 (G249, N273 and N309) were targeted by the NA-reactive antibodies[27], and all these three amino acids are conserved among the N1 neuraminidases between 2.3.4.4b A(H5N1) and recent A(H1N1)pdm09 viruses (Supplementary Fig. 3). Additional

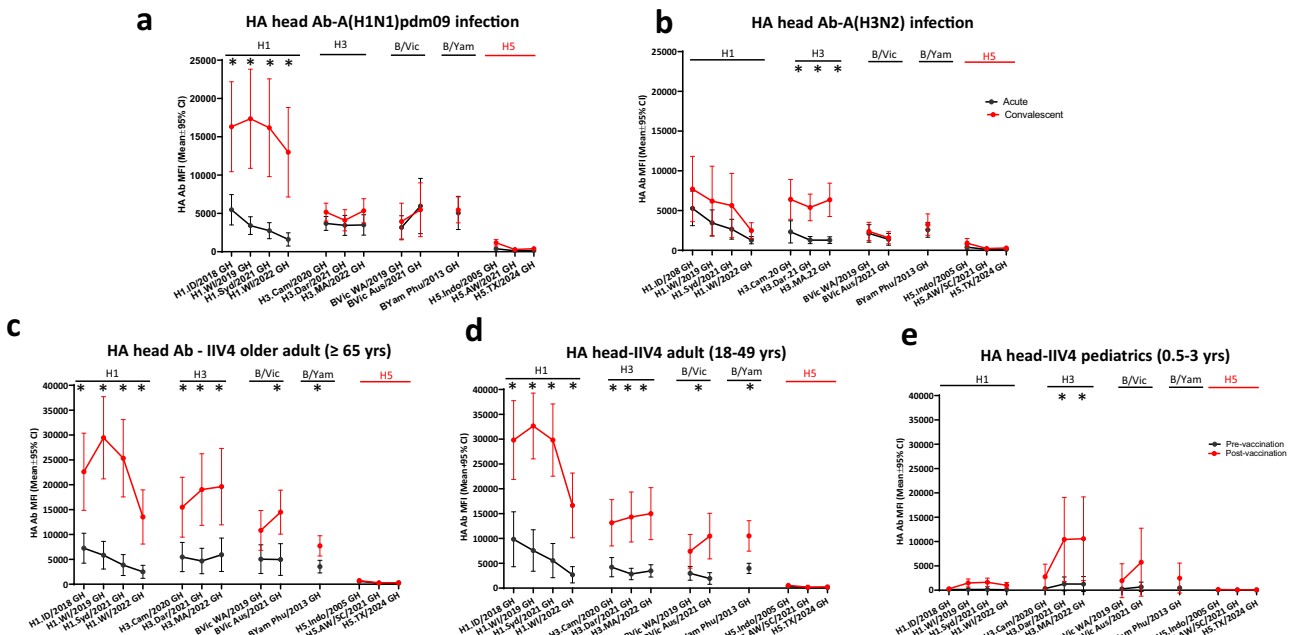

**Fig. 5 | Seasonal Influenza infection and vaccination in 2023–24 did not significantly induce binding antibodies to the HA Head of A(H5) 2.3.4.4b viruses.** Paired serum samples collected from seasonal influenza infection and quadrivalent inactivated influenza vaccination (IIV4) participants during 2023–24 influenza season were tested by 28-Plex MIADA, binding antibody levels (MFI) to HA head were plotted. **a** Acute and convalescent sera from A(H1N1)pdm09 infection ($n = 21$ participants); **b** Acute and convalescent sera from A(H3N2) infection ($n = 13$ participants); (**c**) Pre and post vaccination sera collected from older adults ($\geq 65$ yrs) who received IIV4 ($n = 20$ participants); **d** Pre and post vaccination sera collected from adults (18–49 yrs) who received IIV4 ($n = 20$ participants); **e** Pre and post vaccination sera collected from pediatric participants (0.5–3 yrs) who received IIV4 ($n = 19$ participants). Y axis: Mean MFIs with error bars indicate 95% confidence interval (CI). Statistical analyses were conducted using two tailed paired t-test, $p < 0.05$ is considered statistically significant. *$p < 0.05$.

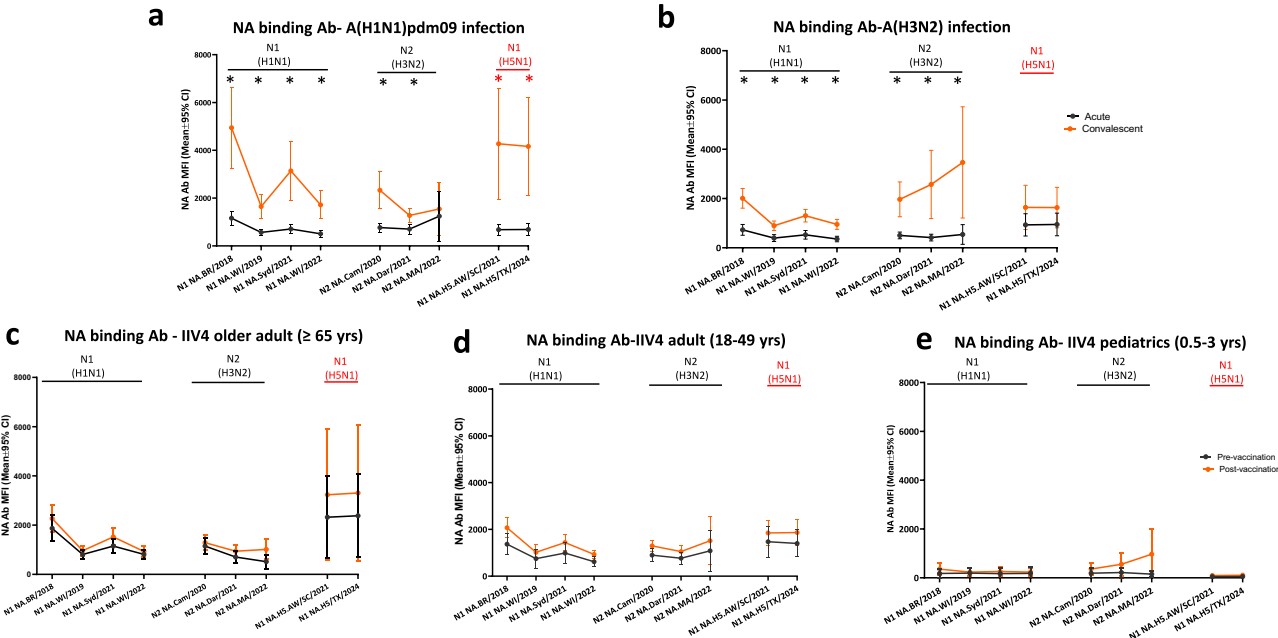

**Fig. 6 | Seasonal A(H1N1)pdm09 influenza virus infection, but not A(H3N2) virus infection or seasonal vaccination induced significant rise of binding antibodies to the N1 neuraminidase of A(H5N1) 2.3.4.4b.** Paired serum samples collected from seasonal influenza infection and quadrivalent inactivated influenza vaccination (IIV4) vaccination participants during 2023–24 influenza season were tested by 28-Plex MIADA, binding antibody levels (MFI) to NA were plotted. **a** Acute and convalescent sera from A(H1N1) infection ($n = 21$ participants); **b** Acute and convalescent sera from A(H3N2) infection ($n = 13$ participants); **c** Pre and post vaccination sera collected from older adults ($\geq 65$ yrs) who received IIV4 ($n = 20$ participants); **d** Pre and post vaccination sera collected from adults (18–49 yrs) who received IIV4 ($n = 20$ participants); **e** Pre and post vaccination sera collected from pediatric participants (0.5–3 yrs) who received IIV4 ($n = 19$ participants). Y axis: mean MFIs with error bars indicate 95% confidence interval (CI). Statistical analyses were conducted using two two-tailed paired t-test, $p < 0.05$ is considered statistically significant, *$p < 0.05$.

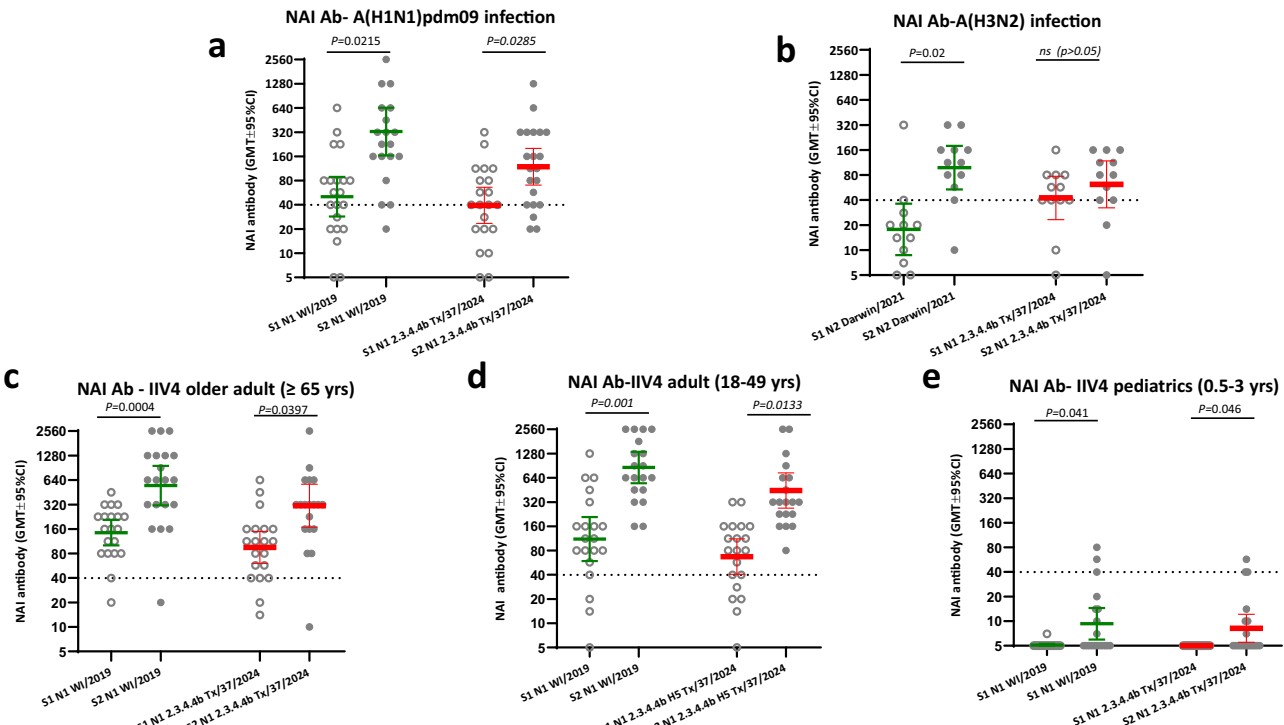

**Fig. 7 | A(H1N1)pdm09 virus infection and seasonal influenza vaccination in 2023–24 induced significant rise of functional neuraminidase inhibition antibody (NAI) responses to N1 of 2.3.4.4b HPAI A(H5N1) virus.** Paired serum samples collected from seasonal influenza A infection and quadrivalent inactivated influenza vaccination (IIV4) vaccination participants during 2023-24 influenza season were tested by ELLA assay using H6Nx reverse genetic viruses with mismatched HA and target NAs. Functional neuraminidase inhibition antibody (NAI) to N1 of 2.3.4.4b A(H5N1) A/Texas/37/2024, N1 from seasonal A(H1N1) and N2 neuraminidase from seasonal A(H3N2) were plotted in scatter plots. **a** NAI antibodies in acute (S1) and convalescent (S2) sera from A(H1N1)pdm09 infection (n = 21 participants); **b** NAI antibodies in acute (S1) and convalescent (S2) sera from A(H3N2) infection (n = 12 participants); **c** NAI antibodies in pre (S1) and post (S2) vaccination sera collected from older adults (≥ 65 yrs) who received IIV4 (n = 20 participants); **d** NAI antibodies in pre (S1) and post (S2) vaccination sera collected from adults (18–49 yrs) who received IIV4 (n = 20 participants); **e** NAI antibodies in pre (S1) and post (S2) vaccination sera collected from pediatric participants (0.5–3 yrs) who received IIV4 (n = 19 participants). Y axis: geometric mean titers (GMT) of NAI antibodies with 95% confidence interval (CI). Statistical comparisons were conducted using two-tailed paired t test, p < 0.05 is considered statistically significant.

NA epitopes, such as S95 and D451 which were shown to play a role in the inhibition of A(H1N1)pdm09 virus[28], were also conserved among the N1 neuraminidases investigated in this study (Supplementary Fig. 3). Past exposure to seasonal A(H1N1)pdm09 viruses could induce cross-reactive functional NAI antibodies to the N1 NA 2.3.4.4b A(H5N1) virus. To date, the threshold of the NAI antibody that correlates with protection still remains to be determined. The amount of neuraminidase in the US-licensed seasonal vaccines is not regulated; thus, the level of cross-reactive NAI antibodies induced by seasonal vaccination may be highly variable. Nevertheless, the role of neuraminidase antibodies in influenza protective immunity has been well recognized[20,29,30]. Whether these pre-existing NAI antibodies could have contributed to the mild clinical outcomes in human A(H5) virus infections detected thus far during the current outbreak warrants further investigation.

Broadly cross-reactive HA stalk antibodies have been considered as one of the targets for universal influenza vaccine because the HA stalk is highly conserved[31,32]. Our results suggested that the group 1 HA stalk antibodies exist in the US population in an age-related pattern: they increased with age and birth cohorts, and can be further boosted upon A(H1N1)pdm09 infection and IIV4 vaccination in adults (Fig. 8). Both A(H1) HA and A(H5) HA belong to the same phylogenetic group 1, and the HA stalks are fairly conserved between the A/Texas/37/2024 H5N1 and A/Michigan/45/2015 H1N1 viruses (Supplementary Fig. 1). Gostic et al. suggested through statistical modeling that childhood imprinting with A(H1N1) virus could provide protection for A(H5N1) viruses (belong to the same phylogenetic group 1) through conserved HA epitopes[33]. Stalk antibodies are thought to offer protection

through Fc receptor-dependent mechanisms such as antibody-dependent cellular cytotoxicity (ADCC). The efficacy of the HA stalk antibodies has been demonstrated in many studies in animal models[34,35], whereas studies in humans that evaluated the protective functions of the HA stalk antibodies thus far have yielded mixed results[23,36–39]. In an A(H1N1)pdm09 human challenge study of healthy volunteers, pre-existing HA stalk antibody correlated with protection in reducing viral shedding, but did not independently predict a decrease in disease severity from influenza infection; rather, neuraminidase antibody correlated with reduction in disease severity[23]. Several recent studies demonstrated the impact of pre-existing immunity from seasonal influenza viruses on the cross-protection against A(H5N1) challenge in the ferret model[13,14,40]. In one study[13], following 2.3.4.4b A(H5N1) virus challenge, ferrets with pre-existing immunity to A(H1N1)pdm09 virus had reduced replication and transmission compared with naive ferrets, supporting our findings that pre-existing A(H1N1)pdm09 immunity may confer some protection against A(H5N1) virus. This study also showed that pre-existing A(H1N1)pdm09 (group 1) immunity in ferrets more effectively reduced the replication and transmission in A(H5N1) virus (group 1) challenge than that in the A(H7N9) virus (group 2) challenge despite the lack of cross-reactive antibodies targeting the HA head, suggesting that the protection was likely mediated by the cross-reactive group 1 HA stalk antibodies and N1 antibodies. Whether and to what extent pre-existing group 1 HA stalk antibodies can confer protection against A(H5N1) virus infection in a clinical setting in humans still needs to be better understood.

Vaccination is the most effective public health measure to prevent influenza, and annual seasonal influenza vaccination is recommended

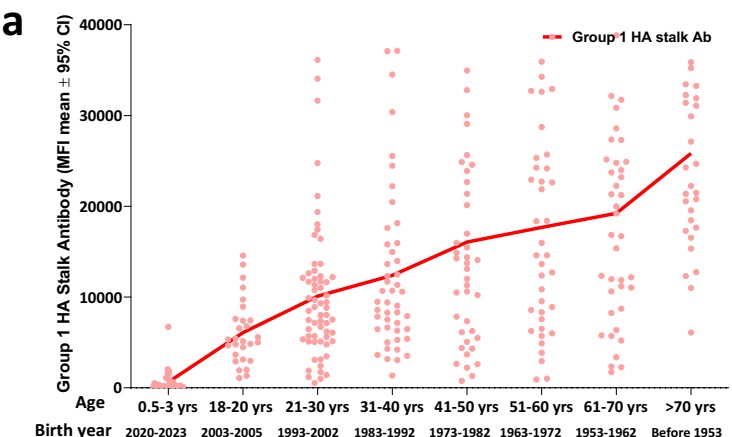

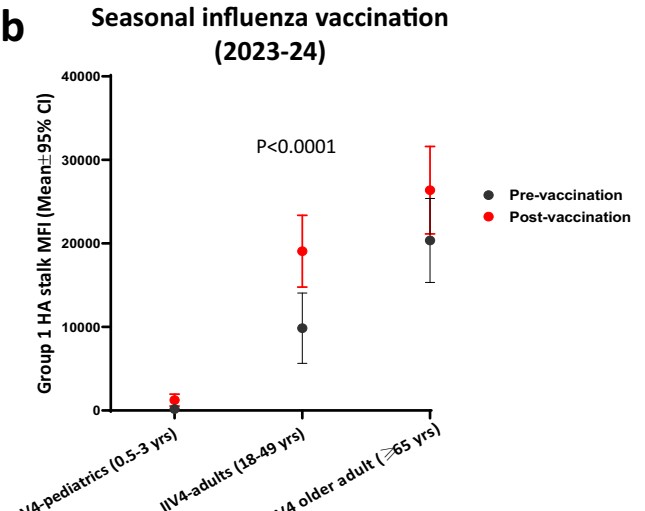

**Fig. 8 | Prevalence of antibodies to the group 1 HA stalk in the US population, and antibody responses to group 1 HA stalk from seasonal influenza virus infection and vaccination. a** Age-stratified scatter plot of antibodies to group 1 HA stalk in 2023–24, 327 sera were collected from 234 participants from 8 age groups. Connected lines represent mean antibody MFI of each age group. **b** Group 1 HA stalk antibodies in pre and post vaccination sera collected in 2023–24 from quadrivalent inactivated influenza vaccination (IIV4) in older adult (≥ 65 yrs)

($n$ = 20 participants), adult (18–49 yrs) ($n$ = 20 participants), and pediatric participants (0.5–3 yrs) ($n$ = 19 participants). **c** Group 1 HA stalk antibodies in acute and convalescent sera from A(H1N1)pdm09 infection ($n$ = 21 participants) and A(H3N2) infection ($n$ = 13 participants) in 2023–24 season. Y axis: mean MFIs with error bars indicate 95% confidence interval (CI). Statistical analyses were conducted using two-tailed paired t test, $p < 0.05$ is considered statistically significant.

for people aged ≥6 months in the US[41]. Influenza vaccines are also anticipated to be used as a medical countermeasure in the event of an influenza pandemic[42]. The egg-based A/gyrfalcon/WA/410886/2014 H5N8 antigen is a part of the U.S. National Pre-Pandemic Influenza Vaccine Stockpile (NPIVS) for pandemic influenza preparedness. This study demonstrated that adjuvanted A/gyrfalcon/Washington/410886/2014 H5N8 vaccine induced both cross-neutralizing and binding antibodies against 2.3.4.4b A(H5N1) viruses (Fig. 2), providing additional data for pandemic preparedness. Our recent study in the ferret model suggested that prior influenza A(H1N1)pdm09 virus infection may confer some protection against influenza A(H5N1) clade 2.3.4.4.b virus[13]. While seasonal influenza vaccines are not intended to prevent A(H5) influenza infection, our data from humans revealed that they could induce cross-reactive and potentially cross-protective NA and HA stalk antibodies; the clinical implications of these responses in A(H5) human infection merit further study.

During a novel influenza A virus outbreak, timely serologic studies can detect both symptomatic and asymptomatic infections and provide more robust estimates of total infections within a

population[43]. When using a single convalescent serum, the seropositivity criteria in the WHO A(H5) case definition[17] requires microneutralization titers ≥40, and either HI ≥ 40, or a positive result from another immunological assay, such as the MIADA assay. We have developed the high-throughput MIADA assay to identify different subtypes of novel influenza A infection[44–46]. In combination with microneutralization assays, MIADA offers a high throughput option for the initial screening during large-scale serosurveys. However, it should be emphasized that due to the complex immunity in the population and cross-reactivity from past exposure to seasonal influenza, binding antibody results must be verified with microneutralization and HI assays using virus antigenically closely related to the viruses that cause the outbreak, and additional serum adsorption may be needed to confirm true seropositivity to A(H5)[45,47,48].

Our study has limitations, first, the clinical protective functions of the cross-reactive neuraminidase and HA stalk antibodies to A(H5N1) viruses in humans need to be further understood; second, although we assessed multiple antibody targets; other adaptive immunity, such as

cell-mediated immunity which can also be associated with reduction of influenza disease severity[16], were not assessed here.

Host immunity is a critical parameter in assessing population susceptibility to influenza virus infections. The Centers for Disease Control and Prevention (CDC) uses the Influenza Risk Assessment Tool (IRAT)[49] to evaluate the pandemic potential for a novel influenza A virus. Population immunity is one of the risk elements used in the IRAT. Our study provides additional data for A(H5N1) pandemic risk assessment. Further studies are needed to better understand the immune responses to A(H5N1) viruses in humans and to what extent pre-existing immunity can deter infection and/or lessen disease severity. Continued surveillance is essential to closely monitor A(H5) viruses for pandemic preparedness.

## Methods

### Human sera
A total of 1868 sera collected from 797 participants in the US were used in the study. Details of the sera collection are described in Table 1.

In an ongoing longitudinal population immunity study, 489 participants from all 10 US HHS regions and 9 age ranges (1–4 yrs, 5–10 yrs, 11–20 yrs, 21–20 yrs, 31–40 yrs, 41–50 yrs, 51–60 yrs, 61–70 yrs, and > 70 yrs) were enrolled in 3 waves from November 2021 to November 2022 (prior to the current A(H5) outbreak in the US) and provided 1467 sera (Table 1 and Fig. 1a, b) for analysis.

Seasonal influenza vaccination, infections, and control sera from the 2023 to 2024 season since the outbreak of 2.3.4.4b A(H5) virus in dairy cattle and poultry, were also evaluated. For influenza vaccination, pre- (S1) and post- (S2) vaccination sera collected through a CDC contract from pediatric (0.5–3 yrs, $n = 19$), adult (18–49 yrs, $n = 20$), and older adult (≥ 65 yrs, $n = 20$) participants who received quadrivalent inactivated influenza vaccines (IIV4) in 2023–24 were analyzed. For seasonal influenza A infection, sera were collected in the US Flu VE network[50] from persons exhibiting acute respiratory illness (ARI) during October 2023 to February 2024 from reverse transcription polymerase chain reaction (RT-PCR) confirmed influenza A(H1N1)pdm09 infection (21 pairs and 41 S1 only), A(H3N2) virus infection (13 pairs and 22 S1 only), and negative cases ($n = 78$) (Table 1). Acute sera were collected from all participants within 7 days post-symptom onset; for those with RT-PCR confirmed A(H1N1)pdm09 or A(H3N2) virus infections, convalescent sera were collected 21–28 days post-symptom onset.

A(H5N8) vaccine sera ($n = 74$) were provided by the Biomedical Advanced Research and Development Authority (BARDA). Stored sera from a BARDA-sponsored vaccine trial (Clinical trial.gov: NCT03497845) were collected from healthy adults (18–49 yrs) who received 2 doses of egg-based influenza A/gyrfalcon/Washington/41088-6/2014 (H5N8) 2.3.4.4c vaccine (CSL Seqirus) with AS03 (GSK) or MF59 (CSL Seqirus) adjuvant (dose 1: day 1, dose 2: day 22) from the U.S. National Pre-Pandemic Influenza Vaccine Stockpile (NPIVS). Sera were collected at day 43 (i.e., 22 days post the second dose). The sample size for this study includes approximately half of the subjects from the per protocol population.

Informed consent was obtained from all participants. For minors, informed consent was obtained from parents/guardians. The use of sera was approved by the Centers for Disease Control and Prevention (CDC) Human Subjects Research Determination. The study was reviewed by CDC and conducted consistent with applicable federal law and CDC policy[51–55].

### Multiplex influenza antibody detection assay (MIADA)
In this study, 12-plex and 28-plex cocktails were used that included influenza HA ectodomain (Ec), HA globular head (GH) from A(H1N1), A(H3N2), A(H5N1), influenza B viruses, and group 1 and group 2 HA stalk; N1 and N2 NAs, influenza A nucleoprotein (NP), and a protein A/G control, details of the antigens used are described in Supplementary

Table 1[56,57]. Antigens were coupled with MagPlex-C Microsphere. Linearity of the antibody levels was first assessed by serial 2-fold diluted curves of representative sera to determine optimal dilution. Then the 1:2000-diluted human sera were used in the assay followed by incubation with the MIADA antigen cocktail on Mini orbital shaker at room temperature. The secondary antibody of phycoerythrin-conjugated goat F(ab')2 anti-human pan Immunoglobulin (PE-pan Ig) reporters were used, and results were analyzed by a Luminex MAGPIX reader to quantify MFIs[58].

### Influenza viruses, molecular and structural analyses
Wild type HPAI 2.3.4.4b A(H5N1) virus isolated from the first dairy cattle outbreak human case A/Texas/37/2024[6] was used in the analyses, additional 2.3.4.4b A(H5N1) virus: A/American wigeon/South Carolina/22-000345-001/2021wild type and candidate vaccine virus (CVV), and wild type A/gyrfalcon/Washington/410886/2014 were also used in the analysis. A(H5) viruses were propagated in 10–11 day old embryonated eggs. Seasonal A(H1N1)pdm09 viruses: A/Wisconsin/67/2022, A/Victoria/2570/2019 and A(H3N2) virus: A/Massachusetts/18/2022 were also used. Seasonal A(H3N2) and A(H1N1) pdm09 viruses were propagated in Madin-Darby Canine Kidney (MDCK)-SIAT1 cells.

All studies using HPAI A(H5Nx) viruses were conducted in Biosafety Level 3 Enhanced (BSL3E) laboratories, studies using seasonal viruses were conducted in Biosafety Level 2 (BSL2) laboratories.

Virus stocks used in the study were sequenced. HA head, stalk, and NA sequences were analyzed using the BioEdit software (https://bioedit.software.informer.com). HA and NA structural models were generated using PyMOL (Molecular Graphics System).

### Hemagglutination Inhibition (HI) assay using horse erythrocytes
HI assays to detect antibody responses to A(H5) viruses were performed with a modified HI assay using horse erythrocytes as previously described[59,60]. In brief, sera were heat-inactivated for 30 min at 56 °C and then tested for non-specific agglutinins and adsorbed with horse erythrocytes as needed. Sera were then treated with receptor-destroying enzyme (RDE) for 18–20 h at 37 °C to remove any non-specific inhibitors that may have been introduced during hemadsorption, followed by 56 °C heat inactivation of RDE for 30 min prior to the HI assay. Sera were serially diluted two-fold and incubated for 30 min with 4 hemagglutination units per 25 μL of virus, and then incubated with 1% horse erythrocytes for 60 min. HI titer was defined as the reciprocal of the last dilution of serum that completely inhibited hemagglutination. Antibody titer <10 (initial dilution) was reported as 5 for statistical analysis.

### Microneutralization (MN) assays
MN assays were performed as previously described in refs. 59–61. Heat-inactivated human sera were serially diluted 2-fold and incubated with one hundred 50% tissue culture infection dose ($TCID_{50}$) of influenza viruses. The virus-serum mixture was used to infect $1.5 \times 10^4$/well MDCK cells and incubated overnight. The plates were fixed with cold 80% acetone and the presence of viral nucleoprotein was quantified by enzyme-linked immunosorbent assay (ELISA). Microneutralization titers were defined as the reciprocal of the highest serum dilution that showed 50% neutralization. Antibody titer <10 (initial dilution) was reported as 5 for statistical analysis.

### Generation of H6Nx reverse genetic viruses with mismatched H6 HA and target NA
Reverse genetics (RG) influenza viruses were generated with a mismatched H6 HA (from A/turkey/Massachusetts/3740/1965, GISAID: EPI407901) and target NA: N1 from 2.3.4.4b A (H5N1) virus A/Texas/37/2024 (GISAID: EPI3171486), N1 from seasonal A(H1N1)pdm09 A/Wisconsin/588/2019 (GISAID: EPI2413538) and N2 from seasonal A(H3N2)

A/Darwin/6/2012 (EPI3009099). Briefly, HA and NA genes of interest and 6 internal genes from A/Puerto Rico/8/1934(H1N1) were synthesized and cloned into the transcription plasmid pCIPolISapIT. Plasmids containing the 8 gene segments were transfected into 293 T cells at 1 μg each using Lipofectamine 2000 to produce recombinant virus. Recombinant virus was propagated in 9-11 day-old embryonated hen's eggs and the genome was confirmed by sequencing. Virus stocks were stored at −80 °C until analysis.

The RG viruses do not have gain of function. Studies using H6Nx RG viruses were conducted in Biosafety II Enhanced Laboratories.

### Enzyme-linked Lectin Assay (ELLA)

Functional neuraminidase inhibition antibodies (NAI) were detected using H6Nx RG viruses by the ELLA assay[62]. Sera were first heat inactivated. Serial two-fold diluted sera were then incubated with H6Nx RG viruses in 96-well plates coated with fetuin for 16–18 h. Horse radish peroxidase (HRP)-labeled peanut agglutinin (lectin) was added to the wells and incubated for 2 h, followed by washing and the addition of substrate to reveal enzymatic cleavage of fetuin by viral NA. NAI titers were calculated as the reciprocal of the highest dilution with at least 50% inhibition of neuraminidase activity. ELLA assays were conducted in Biosafety II Enhanced Laboratories.

### Statistical analysis

To compare the antibody responses, two-tailed Wilcoxon matched-pairs signed rank test (non-parametric) or two-tailed paired t test (parametric) was used, $p < 0.05$ is considered statistically significant. Data was analyzed for normality using Shapiro-Wilk test, Anderson-Darling test, and Kolmogorov-Smirnov test simultaneously. Spearman's rank correlation analysis (two-tailed) was conducted for correlation analysis. Statistical analyses were performed using GraphPad Prism 8.

### Reporting summary

Further information on research design is available in the Nature Portfolio Reporting Summary linked to this article.

## Data availability

All data are shown in the figures, tables and supplementary materials. Source data is provided with this paper. To protect participant privacy, the complete data sets can be obtained by submitting a request to the corresponding author Dr. Min Z. Levine (mlevine@cdc.gov) and will be available to researchers with a methodologically sound proposal. Requests will be reviewed and responded to within 60 days. No custom code was used for this study. Source data are provided with this paper.

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

## Acknowledgements

We thank the investigators from US Flu VE network sites for providing the acute and convalescent sera collected from seasonal influenza infections and influenza-negative controls during 2023–24 sesaon. We thank Ms. Shelly Zimbric, staff from the University of Wisconsin and the study sites of the influenza population immunity study for collecting the 2021–23 population immunity sera. We also thank Xiaonan Sun, Makeda Kay, Sneha Joshi, Hao Zhang and Xiaoyu Zheng from the influenza division, CDC for their assistance. This study is funded by Centers for Disease Control and Prevention (CDC). The collection of population immunity sera was funded under contract: 200-2016-91800 Safety and Healthcare Epidemiology Prevention Research Development (SHEP-heRD). The A(H5N8) vaccine sera were supported with federal funds

from the Biomedical Advanced Research and Development Authority (BARDA), Administration for Strategic Preparedness and Response at the U.S. Department of Health and Human Services, under contracts HHSO100201400003I, HHSO100201600004I, HHSO100201800004I. Opinions, findings, and conclusions reported by the authors are strictly their own and are in no way meant to represent the opinion, views, policies or the official position of CDC, BARDA and U.S. Department of Health and Human Services.

## Author contributions

M.Z.L. conceived the study; M.Z.L. and Z.L. designed the study; Z.L., F.L., Y.J., S.J., C.H., F.L.G., W.T., P. C., I.Y., C.T.D., J.S., T.T. acquired the data. N.S., A.K., S. Z., J.Z., M.M., V.C., C.O. provided sera. M.Z.L. analyzed the data and interpreted the results. M.Z.L., Z.L., F.L. wrote the manuscript. M.Z.L. supervised the study. All co-authors were involved in the manuscript preparation process for important intellectual content.

## Competing interests

The authors declare no competing interests.
