## [Transparent Peer Review file · Nature Communications]

Pre-existing Cross-reactive Immunity to Highly Pathogenic Avian Influenza 2.3.4.4b A(H5N1) Virus in the United States

Corresponding Author: Dr Min Levine

Version 0:

Reviewer comments:

Reviewer #1

(Remarks to the Author)

In this study, the authors aim to determine the impact of pre-existing cross-reactive immunity to hemagglutinin head, stalk, and neuraminidase from seasonal influenza against H5N1.

The abstract states that pre-existing neutralizing and hemagglutinin (HA)-head-binding antibodies to H5N1 were "low". However, this appears to be at odds with the data in the manuscript: On page 5, line 111 "All sera were negative (<10) against both wild type A(H5N1) viruses in both assays (Table 2), confirming no seropositivity among 1467 US population immunity sera surveyed across 3 time points during 2021-23 seasons (Fig 1)". This negative data does not support the position that seasonal or pandemic influenza will provide cross-protection against H5N1.

The abstract states that antibodies to group 1 HA stalk were prevalent. However, there is currently no clinical data to support a protective role for anti-HA stalk immunity. There are at least 5 clinical trials that indicate HA stalk antibodies play no significant role in protection against influenza illness/disease.

Examples:

Reference # 20: Park et al., mBIO 2018 shows that natural pre-existing anti-HA stalk antibodies correlated with reduced viral shedding after direct human challenge with influenza but did not decrease disease severity.

Reference #31: Han et al. Inf Dis Soc America 2021 shows that administration of anti-HA stalk antibody (MAb: CR6261) does not reduce disease severity after direct human challenge with influenza

Reference #32: Lim et al., Antimicrob Agents Chemother 2020 shows that administration of anti-HA stalk antibody (MAb: MHAA4549A) does not reduce disease severity after direct human challenge with influenza

Reference #33: Lim et al., OFID 2022 shows that administration of anti-HA stalk antibody (MAb MHAA4549A) does not reduce disease severity among influenza-infected outpatients.

S. K. Tan et al., A Randomized, Placebo-Controlled Trial to Evaluate the Safety and Efficacy of VIR-2482 in Healthy Adults for Prevention of Influenza A Illness (PENINSULA). Clin Inf Dis 2024;79:1054. This study of 2,977 participants shows that administration of an anti-HA stalk antibody (MAb: VIR-2482) did not significantly prevent naturally occurring protocol-defined PCR-confirmed influenza illness.

This series of clinical trials represent an impressive dataset in the sense that they show that a) natural levels of pre-existing anti-HA stalk immunity do not correlate with protection against influenza disease symptoms (Park 2018), b) administration of high concentrations of monoclonal anti-HA stalk antibodies fail to protect against direct human influenza challenge (Han 2021, Lim 2020) and c) anti-HA stalk Mabs fail to protect against natural influenza infection (Lim 2022, Tan 2024). Bearing this in mind, it is unlikely that anti-HA stalk antibodies described in this current paper will have an impact on H5N1 infection.

The authors identified cross-reactive anti-NA immunity (NAI), which is to be expected from H1N1 cross-reactivity to the neuraminidase of H5N1. However, it is unclear what level of NAI is necessary to provide an impact on H5N1 disease severity. Without a correlate of cross-reactive immunity, the reader is unable to determine if 1%, 10% or 100% of the population would be expected to have NAI-mediated protection against H5N1. For this reason, the significance of the study is lacking since the results appear to be largely descriptive in nature.

The paper also includes some data from an H5N1 vaccination trial. However, it is unclear why it is included here since it does not provide information on the role of pre-existing population immunity to H5N1 as indicated by the title of the paper. Moreover, this H5N1 vaccine-based dataset appears to be incomplete – based on responses to prior reviewer comments, the authors did not have enough serum to perform H5N1 neutralizing assays (ferret serum was substituted for human serum for positive and negative controls in Table 2) and the authors are missing key information regarding how the participants were vaccinated, including no information on which individuals received H5N1 vaccination with AS03 adjuvant and which ones were vaccinated with H5N1 vaccine formulated with MF59 adjuvant.

Version 1:

Reviewer comments:

Reviewer #4

(Remarks to the Author)

Li et al. collected serum samples from human cohorts across 10 regions of the United States over three consecutive influenza seasons. Using these samples, they evaluated antibody responses to circulating human influenza viruses and two clade 2.3.4.4b H5N1 viruses, including a recent dairy cow isolate. During the final influenza season, they also monitored patients for influenza infections, collecting blood samples at the time of confirmed infection and after recovery. Additionally, they assessed antibody responses before and after seasonal influenza vaccination during this last season. Finally, the authors evaluated the cross-reactivity of human antisera from a vaccine trial using a 2.3.4.4c H5 hemagglutinin vaccine.

From the U.S. cohorts, they found that patients had neutralizing antibodies against H1 HA, but little to no neutralizing activity against H5 HA. In contrast, patients with immunity to H1N1 displayed cross-reactive antibodies against the N1 of H5N1 viruses. Further analysis revealed that most individuals had antibodies targeting the HA stalk of both H1 and H5, with prevalence increasing with age. Lastly, sera from individuals vaccinated with the clade 2.3.4.4c H5 hemagglutinin vaccine neutralized the clade 2.3.4.4b viruses.

Major Comments:

1) Throughout the manuscript the authors indicate that humans are not immunologically naïve to H5N1 and that humans have substantial immunity towards H5N1 viruses. This is not accurate, and the findings have not been placed in the appropriate context. The human population has extensive pre-existing immunity to seasonal H1N1 and H3N2 viruses, and the author's findings and those of others show that there are cross-reactive antibodies towards the HA stalk between H1 and H5, and towards the N1 of human H1N1 viruses and the N1 of clade 2.3.4.4b H5N1 viruses. The authors instead present the findings in the reverse context indicating that humans have pre-existing immunity to H5N1 and that this is the result of seasonal virus infection or exposure. It is inaccurate to claim that humans are not immunologically naïve to H5N1 because they do not have neutralizing antibodies towards the H5 HA, and all the antibodies detected are cross reactive with H1N1 and not H5 specific. It would be more accurate to instead present the data showing the antibody response to seasonal influenza viruses and then outline the cross-reactivity of this response towards H5N1 viruses. The data is scientifically sound and provides a unique data set describing immunity across the US, but the data needs to be presented in a more accurate context.

2) There are factual errors, inconsistencies, and inaccurate terminology. For example, in line 53, the authors indicate that an influenza pandemic occurs when "highly pathogenic novel influenza viruses mutate to gain sustained transmissibility". Yet, a highly pathogenic virus has not caused a pandemic in humans. Line 54, indicates "host immunity is critical for assessing susceptibility to influenza virus infection....." This would be more accurate to indicate "pre-existing host immunity" and susceptibility to "avian or emerging influenza virus infection". Similarly, in lines 45-52, the authors do not clearly separate clade 2.3.4.4b H5N1 infections in humans prior to the bovine H5N1 outbreak and then afterwards. Lines 68-71 indicate "We analyzed whether seasonal influenza A virus infection or seasonal influenza vaccination can induce cross-reactive antibodies to 2.3.4.4b A(H5N1) viruses". This is not entirely accurate. Vaccination and infection are inducing antibodies against the infecting virus or the vaccine. These antibodies happen to be cross-reactive to H5N1, but the vaccine or infection is not necessarily inducing cross-reactive antibodies. The manuscript needs to be revised throughout for accuracy in terminology.

3) It is unclear from the statistical analysis section if the data was tested for normality. Please indicate if normality was tested, and if normality was not tested or the data are not normally distributed, please using non-parametric statistical tests. Please add statistical analyses to Figures 1c-e and 2a-d.

Minor Comments:

1) Lines 105-114. This section of the text provides the WHO definition of seropositivity to H5N1. The authors indicate that they tested 12 samples that were cross-reactive to H5N1 viruses by MFI to confirm seropositivity. The MFI measures binding

- antibodies and humans can develop cross-reactive HA head antibodies. Given that H5N1 viruses are not circulating in humans, it more likely that patients in the cohort have cross reactive HA head binding antibodies than they were infected with H5N1. Thus, the authors are actually confirming seronegativity. Please revise this section for accuracy.
- 2) Line 148 indicates “We analyzed the prevalence of antibodies towards the N1 neuraminidase of two 2.3.4.4b viruses”. It would be more accurate to state “cross-reactive antibodies”.
 - 3) Line 184 indicates “to elucidate the source of the anti HA stalk antibodies”. The approach used does not “elucidate” the source. Please revise.
 - 4) Line 197-198 indicates the “study provides a timely and comprehensive assessment of population immunity in the US to 2.3.4.4b A(H5N1) viruses.” This is not accurate. Instead, the study provides an assessment of cross-reactive immunity present in humans.
 - 5) Line 220, please revise for accuracy. This sentence indicates vaccination, or infection confers prophylactic protection against H5N1 infection in mice, ferrets, and humans. This is not known for humans. Please revise.
 - 6) Line 233, please provide references indicating NA immunity is protective.
 - 7) Line 249, several ferret studies have been published evaluating cross protective immunity to seasonal viruses against H5N1. Please include these references.
 - 8) Lines 258-268. Please revise this section to clarify that vaccination prevents influenza illness and disease, and not infection.
 - 9) Table 2. It would strengthen the dataset to add neutralizing titers in the human serum against a circulating H1N1 virus.

Version 2:

Reviewer comments:

Reviewer #4

(Remarks to the Author)

Li et al. have submitted a revised version of their manuscript evaluating cross-reactive population wide immunity to clade 2.3.4.4b H5N1 viruses. The revised manuscript is much improved, but there are still some comments to address.

Comments:

- 1) The title and throughout the manuscript use the term “Population Immunity”. This is the term used in IRAT. However, this terminology is not accurate, and the studies in the manuscript evaluate cross-reactive antibodies present in human serum. There is no epidemiological evidence that H5N1 viruses are causing routine infections in humans, and the term “population immunity” in risk assessment tools is meant to capture the degree to which pre-existing immunity to seasonal influenza viruses cross-reacts with emerging viruses. Thus, this term should be replaced. It would be more accurate to state “cross-reactive antibodies”, or “pre-existing cross-reactive antibodies”. This will be more accurate and avoid confusion for readers outside the influenza field.
- 2) Line 70 should be revised. Do the authors have another explanation for why humans have population wide immunity to H1N1 and H3N2 viruses and/or how there would be population wide immunity the H5N1 viruses? It would be more accurate to re-phrase the text to indicate the authors sought to confirm or validate that prior vaccination or infection was inducing cross-reactive antibodies.
- 3) The introduction is missing required background. Please add a description of seasonal influenza viruses and the immunity that is induced by infection with these viruses and vaccination.
- 4) Lines 105-116. This section is written to indicate that antibody analyses are being conducted to confirm “seropositivity”. This is misleading. The authors are confirming seronegativity. There is no evidence that H5N1 is circulating in humans or causing routine undetected infections. While the MIADA assay may yield a positive result, this is expected as ELISA assays by many other groups also show cross-reactive antibodies between H1 HA and H5 HA. This needs to be revised and phrased to indicate the studies are being performed to confirm seronegativity as there is no evidence of on-going H5N1 infections in humans outside of farms.
- 5) This comment was in the previous review and still needs to be addressed. It is unclear from the statistical analysis section if the data was tested for normality. In the response to reviewers, the authors indicate that normality was tested. Please update the statistical analysis section to include these details and the test used, and please also update this section to include other tests that were performed such as correlations etc.
- 6) In Figures 6 and 7 the results are contradictory. Figure 6c-e shows that upon vaccination there is no significant increase in cross-reactive N1 binding antibody titers measured via MIADA. However, in Fig 7c-e, there is a significant increase in N1 inhibitory antibody titers upon vaccination. The authors need to address this discrepancy in the discussion. Alternatively, MIADA assays need to be performed at a lower starting dilution to capture differences in antibody titers.
- 7) There are still sections that need to be modified for proper grammar and structure. For example, there are 1 sentence paragraphs throughout the manuscript. Both the last sentence of the abstract and discussion are incomplete and/or not required. Line 178 indicates HA stalks are cross-reactive. Antibodies to the stalks are cross-reactive. Please revised. Line

211, please change “rise” to “increase” or similar.

8) Table 2. The notes indicate titers less than 10 are reported as 5. However, HI antibody titers are reported as 7. Please revise.

NCOMMS-25-41173-T

Population Immunity to Hemagglutinin Head, Stalk and Neuraminidase of Highly Pathogenic Avian Influenza 2.3.4.4b A(H5N1) viruses in the United States and the Impact of Seasonal Influenza on A(H5N1) Immunity

Reviewers' comments:

Reviewer #1 (Remarks to the Author):

In this study, the authors aim to determine the impact of pre-existing cross-reactive immunity to hemagglutinin head, stalk, and neuraminidase from seasonal influenza against H5N1.

Response:

Amid the unprecedented outbreak of the Highly pathogenic avian influenza (HPAI) 2.3.4.4b A(H5N1) viruses in dairy cattle and poultry and spilled over to humans, the main objective of our study is to provide a comprehensive assessment of the population immunity in the United States (US) by measuring the levels of antibodies to multiple immunological markers (hemagglutinin head, stalk, and neuraminidase) that could contribute to protection against 2.3.4.4b A(H5N1) infection in the US population. We have demonstrated the impact of the pre-existing immunity against 2.3.4.4b A(H5N1) infection in an earlier study (ref 13, Sun et al).

The abstract states that pre-existing neutralizing and hemagglutinin (HA)-head-binding antibodies to H5N1 were “low”. However, this appears to be at odds with the data in the manuscript: On page 5, line 111 “All sera were negative (<10) against both wild type A(H5N1) viruses in both assays (Table 2), confirming no seropositivity among 1467 US population immunity sera surveyed across 3 time points during 2021-23 seasons (Fig 1)”. This negative data does not support the position that seasonal or pandemic influenza will provide cross-protection against H5N1.

Response:

Thanks for the comments.

There are two separate assessments here: seropositivity vs protective immunity, these overlap but not the same.

- **Seropositivity to A(H5):** *Based on WHO criteria, A(H5) seropositivity is defined by MN/ HI (or H5 positive binding antibody) which mostly measure antibodies targeting the HA head only. These antibodies are subtype (H5)-specific thus can be a good indicators of A(H5N1) infection for seroprevalence studies. What our results*

showed with no A(H5) seropositivity indicated that there was no prior A(H5N1) infection in the US population. However, MN/HA and HA head binding antibodies are only one arm of the protective immunity against A(H5N1).

- **Protective immunity against A(H5):** Multiple immune mechanisms could contribute to protection against influenza infection, including MN, HI antibodies, and antibodies targeting HA head, HA stalk and NA antibodies. The objective of our study is to provide a comprehensive assessment of all these immunological markers that could contribute to protection against A(H5N1) infection in the US population.

As demonstrated by several recent studies published in 2025 that specifically assessed the impact of pre-existing immunity on protective immunity against 2.3.4.4.b A(H5N1), including A(H5N1) ferret challenge study conducted by our group, other recent reports in the ferret challenge model and study in humans (references below and ref 13-15 of the manuscript), these new data specific to 2.3.4.4b A(H5N1) suggest that pre-existing immunity could offer some protection against A(H5N1) infection mediated through NA antibodies and/or HA stalk antibodies.

Sun et al., EID 2025 March 31(3):458-466. Effect of Prior Influenza A(H1N1)pdm09 Virus Infection on Pathogenesis and Transmission of Human Influenza A(H5N1) Clade 2.3.4.4b Virus in Ferret Model - PubMed

Le Sage et al., EID 2025 March 31 (3) 477-487: Influenza A(H5N1) Immune Response among Ferrets with Influenza A(H1N1)pdm09 Immunity - PubMed

Garretson et al., Nature Medicine 31 1454-1458 (2025); Immune history shapes human antibody responses to H5N1 influenza viruses | Nature Medicine

Non-seropositive (defined based on negative MN/HA) sera could also have cross-reactive NA antibodies and HA stalk antibodies, multiple immune mechanisms (MN, HI, HA stalk and NA antibodies etc) could contribute to the protective immunity against influenza A(H5N1) infection.

The abstract states that antibodies to group 1 HA stalk were prevalent. However, there is currently no clinical data to support a protective role for anti-HA stalk immunity. There are at least 5 clinical trials that indicate HA stalk antibodies play no significant role in

protection against influenza illness/disease.

Examples:

Reference # 20: Park et al., mBIO 2018 shows that natural pre-existing anti-HA stalk antibodies correlated with reduced viral shedding after direct human challenge with influenza but did not decrease disease severity.

Reference #31: Han et al. Inf Dis Soc America 2021 shows that administration of anti-HA stalk antibody (MAb: CR6261) does not reduce disease severity after direct human challenge with influenza

Reference #32: Lim et al., Antimicrob Agents Chemother 2020 shows that administration of anti-HA stalk antibody (MAb: MHAA4549A) does not reduce disease severity after direct human challenge with influenza

Reference #33: Lim et al., OFID 2022 shows that administration of anti-HA stalk antibody (MAb MHAA4549A) does not reduce disease severity among influenza-infected outpatients.

S. K. Tan et al., A Randomized, Placebo-Controlled Trial to Evaluate the Safety and Efficacy of VIR-2482 in Healthy Adults for Prevention of Influenza A Illness (PENINSULA). Clin Inf Dis 2024;79:1054. This study of 2,977 participants shows that administration of an anti-HA stalk antibody (MAb: VIR-2482) did not significantly prevent naturally occurring protocol-defined PCR-confirmed influenza illness.

This series of clinical trials represent an impressive dataset in the sense that they show that a) natural levels of pre-existing anti-HA stalk immunity do not correlate with protection against influenza disease symptoms (Park 2018), b) administration of high concentrations of monoclonal anti-HA stalk antibodies fail to protect against direct human influenza challenge (Han 2021, Lim 2020) and c) anti-HA stalk Mabs fail to protect against natural influenza infection (Lim 2022, Tan 2024). Bearing this in mind, it is unlikely that anti-HA stalk antibodies described in this current paper will have an impact on H5N1 infection.

Response:

Thanks for providing the additional references. However, these references are based on data against seasonal influenza, not to A(H5N1). In addition, in the first example from the reviewer's list: Park et al. reported that higher pre-challenge anti-stalk antibody titers significantly correlated with shorter duration of virus shedding, less number of symptoms, and less presence of mild to moderate influenza disease (Park 2018). This study indicated that anti-stalk antibodies could at least in part, play a role in mitigating disease severity and also possibly limiting the virus transmission.

To investigate the impact of pre-existing immunity against 2.3.4.4b A(H5N1) infection from the current outbreak, we did perform 2.3.4.4b A(H5N1) challenge in the ferret model (ref 30 Sun, X., et al. Effect of Prior Influenza A(H1N1)pdm09 Virus Infection on Pathogenesis and Transmission of Human Influenza A(H5N1) Clade 2.3.4.4b Virus in Ferret Model EID 458-466, 2025), and showed that ferrets with prior immunity to A(H1N1)pdm09 virus (group 1) had reduced replication and transmission following 2.3.4.4b H5N1 virus challenge compared with naive ferrets, and these protection were likely mediated by HA stalk and NA antibodies, not antibodies to the HA head. Finding from our studies are supportive of what was reported by Park et al. (Park 2018).

Multiple immune mechanisms could contribute to protection against influenza A(H5N1). As demonstrated by several recent studies published in 2025 that specifically assessed the impact of pre-existing immunity on protective immunity against 2.3.4.4b A(H5N1), including A(H5N1) ferret challenge study conducted by our group, other recent reports in the ferret challenge model and study in humans (references below), these new data specific to 2.3.4.4b A(H5N1) shed new lights on the potential impact of the pre-existing immunity on A(H5N1) infection, suggesting that pre-existing immunity could potentially offer some protection against A(H5N1) infection mediated likely through NA antibodies and/or HA stalk antibodies. We have included these new data and references in the revised manuscript.

Sun et al., EID 2025 March 31(3):458-466. Effect of Prior Influenza A(H1N1)pdm09 Virus Infection on Pathogenesis and Transmission of Human Influenza A(H5N1) Clade 2.3.4.4b Virus in Ferret Model - PubMed

Le Sage et al., EID 2025 March 31 (3) 477-487: Influenza A(H5N1) Immune Response among Ferrets with Influenza A(H1N1)pdm09 Immunity - PubMed

Garretson et al., Nature Medicine 31 1454-1458 (2025); Immune history shapes human antibody responses to H5N1 influenza viruses | Nature Medicine

The authors identified cross-reactive anti-NA immunity (NAI), which is to be expected from H1N1 cross-reactivity to the neuraminidase of H5N1. However, it is unclear what level of NAI is necessary to provide an impact on H5N1 disease severity. Without a correlate of cross-reactive immunity, the reader is unable to determine if 1%, 10% or 100% of the population would be expected to have NAI-mediated protection against H5N1. For this reason, the significance of the study is lacking since the results appear to be largely descriptive in nature.

Response:

The main goal of our current study is to assess multiple arms of pre-existing immunity to the 2.3.4.4b H5N1 virus in US population in the context of current H5 outbreak. We found that there are pre-existing cross-reactive antibodies to N1 of 2.3.4.4b A(H5N1) in the population with an age-related pattern. Although to date, the threshold of the NA antibody that correlates with protection in the clinical setting still remains to be determined, the protective functions of the NA antibodies were demonstrated in many studies and well documented in influenza literature. Our study found that certain age groups in the US population have quite high levels of pre-existing NAI antibodies to N1 of A(H5N1), it is plausible that these pre-existing NAI antibodies could provide some levels of protection, this is an important finding to inform the pandemic risk assessment of these emerging A(H5N1) viruses, especially given the mild clinical symptoms presented in most cases from the current outbreak.

The paper also includes some data from an H5N1 vaccination trial. However, it is unclear why it is included here since it does not provide information on the role of pre-existing population immunity to H5N1 as indicated by the title of the paper. Moreover, this H5N1 vaccine-based dataset appears to be incomplete – based on responses to prior reviewer comments, the authors did not have enough serum to perform H5N1 neutralizing assays (ferret serum was substituted for human serum for positive and negative controls in Table 2) and the authors are missing key information regarding how the participants were vaccinated, including no information on which individuals received H5N1 vaccination with AS03 adjuvant and which ones were vaccinated with H5N1 vaccine formulated with MF59 adjuvant.

Response:

The data from A(H5N1) vaccines trial is not only to demonstrate the current available 2.3.4.4c vaccine can induced cross-reactive neutralizing antibodies to the 2.3.4.4b viruses, it also serves as critical positive control sera with cross-reactivity to 2.3.4.4b viruses to define MIADA assay sensitivity, specificity and thresholds as listed in table S2.

Manuscript NCOMMS-25-41173-A-Z

Point-by-point response to reviewer comments:

Reviewer #4 (Remarks to the Author):

Li et al. collected serum samples from human cohorts across 10 regions of the United States over three consecutive influenza seasons. Using these samples, they evaluated antibody responses to circulating human influenza viruses and two clade 2.3.4.4b H5N1 viruses, including a recent dairy cow isolate. During the final influenza season, they also monitored patients for influenza infections, collecting blood samples at the time of confirmed infection and after recovery. Additionally, they assessed antibody responses before and after seasonal influenza vaccination during this last season. Finally, the authors evaluated the cross-reactivity of human antisera from a vaccine trial using a 2.3.4.4c H5 hemagglutinin vaccine.

From the U.S. cohorts, they found that patients had neutralizing antibodies against H1 HA, but little to no neutralizing activity against H5 HA. In contrast, patients with immunity to H1N1 displayed cross-reactive antibodies against the N1 of H5N1 viruses. Further analysis revealed that most individuals had antibodies targeting the HA stalk of both H1 and H5, with prevalence increasing with age. Lastly, sera from individuals vaccinated with the clade 2.3.4.4c H5 hemagglutinin vaccine neutralized the clade 2.3.4.4b viruses.

Major Comments:

1) Throughout the manuscript the authors indicate that humans are not immunologically naïve to H5N1 and that humans have substantial immunity towards H5N1 viruses. This is not accurate, and the findings have not been placed in the appropriate context. The human population has extensive pre-existing immunity to seasonal H1N1 and H3N2 viruses, and the author's findings and those of others show that there are cross-reactive antibodies towards the HA stalk between H1 and H5, and towards the N1 of human H1N1 viruses and the N1 of clade 2.3.4.4b H5N1 viruses. The authors instead present the findings in the reverse context indicating that humans have pre-existing immunity to H5N1 and that this is the result of seasonal virus infection or exposure. It is inaccurate to claim that humans are not immunologically naïve to H5N1 because they do not have neutralizing antibodies towards the H5 HA, and all the antibodies detected are cross reactive with H1N1 and not H5 specific. It would be more accurate to instead present the data showing the antibody response to seasonal influenza viruses and then outline the cross-reactivity of this response towards H5N1 viruses. The data is

scientifically sound and provides a unique data set describing immunity across the US, but the data needs to be presented in a more accurate context.

Response:

Thanks for the comments. We have made edits throughout the manuscript to improve clarity on the points raised by the reviewer.

Specifically, per reviewer's comments, we deleted the sentence that states: "the population in the US was not completely immunologically naïve to the 2.3.4.4.b virus" in the revised version (as shown in line 208 of track changed revision). In addition, we further specified that the neuraminidase antibodies and HA stalk antibodies detected are cross-reactive antibodies from seasonal influenza viruses and not from infection with A(H5N1) viruses.

To further clarify, amid the unprecedented outbreak of the A(H5N1) viruses in humans in the US, our study was indeed initiated to first to assess the levels of "population immunity" to the novel 2.3.4.4b A(H5N1) viruses in the US to for pandemic risk assessment, as how is presented in the manuscript. Population immunity is a critical element of influenza risk assessment, as outlined in CDC's influenza risk assessment tool (IRAT) Influenza Risk Assessment Tool (IRAT) | Pandemic Flu | CDC (line 55 and reference 11). For novel influenza viruses of which there have not been wide circulations in the human population, pre-existing immunity against these novel viruses, if there are any, are often derived from cross-reactivity from past exposure to seasonal influenza viruses. Then, from the analysis of the population immunity sera, we determined none of the participants had serologic evidence of A(H5N1) infection however there were substantial levels of cross-reactive NA and HA stalk antibodies to A(H5N1), we then set out to investigate whether these cross-reactive antibodies were derived from seasonal Influenza A(H1N1)pdm09 vs A(H3N2), and whether they were from seasonal influenza infection vs vaccination. This is how the studies were carried out and thus how the results are presented.

Multiple immune mechanisms can contribute to the protective immunity against influenza. Here, to provide a comprehensive assessment of immunity, we analyzed multiple immunological markers that could contribute to protection against A(H5N1) and contribute to the population's susceptibility to infection and severity of the illness. We not only assessed the levels of antibodies targeting the HA head, such as neutralizing antibodies, we also assessed the levels of neuraminidase antibodies and HA stalk antibodies. Neutralizing antibodies and antibodies targeting the HA head are often subtype-specific, the presence of neutralizing and antibodies specific to A(H5N1) are indicative of serological evidence of A(H5N1) infection, therefore these are the targets used in WHO criteria to define 'seropositivity' for A(H5N1) case definition of human infection: [2](https://www.who.int/teams/global-influenza-programme/avian-influenza/case-<div data-bbox=)

definitions. Our study found that there were no pre-existing neutralizing and antibodies targeting the A(H5N1) HA head, confirming there is no serologic evidence of infection in the participants surveyed. However, we found there are pre-existing cross-reactive antibodies targeting the neuraminidase and HA stalk of the A(H5N1) viruses: although these antibodies are not A(H5) specific, they can inhibit the function of neuraminidase from 2.3.4.4b H5N1 virus, and bind to group 1 HA stalk, could potentially provide protective immunity against the 2.3.4.4b H5N1 virus that causing the outbreak in the US.

2) There are factual errors, inconsistencies, and inaccurate terminology. For example, in line 53, the authors indicate that an influenza pandemic occurs when “highly pathogenic novel influenza viruses mutate to gain sustained transmissibility”. Yet, a highly pathogenic virus has not caused a pandemic in humans

Response: We have now revised this sentence to below (Line 53-54 in track changed version):

“Influenza pandemics can occur when novel influenza viruses that can infect humans mutate to gain sustained transmissibility among populations that have little or no pre-existing immunity.”

Line 54, indicates “host immunity is critical for assessing susceptibility to influenza virus infection.....” This would be more accurate to indicate “pre-existing host immunity” and susceptibility to “avian or emerging influenza virus infection”.

Response: we have revised this sentence as suggested by the reviewer (Line 54-56 in track changed version):

“Pre-existing host immunity is a critical parameter in assessing susceptibility to novel avian and other emerging influenza virus infections and the risk of an influenza pandemic.”

Similarly, in lines 45-52, the authors do not clearly separate clade 2.3.4.4b H5N1 infections in humans prior to the bovine H5N1 outbreak and then afterwards

Response:

We have clearly separated the A(H5N1) human cases in the US prior to the bovine outbreak vs those since the bovine A(H5N1) outbreak:

- Prior to the current bovine A(H5N1) outbreak, there was only one US human A(H5N1) case in the US, that is described in line 43-44:

“The first A(H5N1) human case in the US was identified in 2022 from direct exposure to 2.3.4.4b H5 infected poultry”

- Since the bovine A(H5N1) outbreak from March 2024, there are 70 reported cases. The detailed summary of the human cases caused by the bovine outbreak since March 2024 is described in line 45-52:

“Since March 2024, an unprecedented outbreak has been ongoing among US dairy cattle in multiple states caused by 2.3.4.4b HPAI A(H5N1) viruses⁶. These viruses have transmitted among cattle, from cattle back to poultry, and spilled over into humans⁶⁻⁸. In March 2024, A(H5N1) human infection was identified in a Texas dairy farm worker after exposure to presumably infected dairy cattle, a HPAI 2.3.4.4b H5N1 virus was isolated, marking the first known case of cattle-to-human transmission of an avian influenza A virus^{6,9}. As of Sept 10, 2025, 70 A(H5N1) human cases have been confirmed in multiple US states from the current outbreak. Most (65/70) were associated with exposure to infected dairy cattle or poultry, and all were caused by clade 2.3.4.4b HPAI A(H5N1) viruses¹⁰, raising concerns of the pandemic risk of these viruses¹¹”

Lines 68-71 indicate “We analyzed whether seasonal influenza A virus infection or seasonal influenza vaccination can induce cross-reactive antibodies to 2.3.4.4b A(H5N1) viruses”. This is not entirely accurate. Vaccination and infection are inducing antibodies against the infecting virus or the vaccine. These antibodies happen to be cross-reactive to H5N1, but the vaccine or infection is not necessarily inducing cross-reactive antibodies. The manuscript needs to be revised throughout for accuracy in terminology.

Response:

Thanks for the comments. Based on the reviewer’s suggestion, we revised this section to improve accuracy and clarity, it now reads (Line 69-72 in track changed version):

“We analyzed whether antibodies induced by seasonal influenza A virus infection or seasonal influenza vaccination are cross-reactive to 2.3.4.4b A(H5N1) viruses to investigate the plausible sources of the pre-existing cross-reactive immunity to A(H5N1) viruses.”

3) It is unclear from the statistical analysis section if the data was tested for normality. Please indicate if normality was tested, and if normality was not tested or the data are not normally distributed, please using non-parametric statistical tests. Please add statistical analyses to Figures 1c-e and 2a-d.

Response:

We have tested normality of the data distribution and updated the statistical analyses in Figure 1 and 2 accordingly.

Figure 1c-e: Wilcoxon matched-pairs signed rank test was performed to compare the MFI values between Wis/2019 H1 and AW/2021 H5 HA head per age group in each wave.

Figure 2a: Wilcoxon matched-pairs signed rank test was performed to compare the MN titers between GF/WA/2014 and AW/2021 viruses.

Figure 2b-d: Spearman's rank correlation analyses were used for correlation analysis.

Minor Comments:

1) Lines 105-114. This section of the text provides the WHO definition of seropositivity to H5N1. The authors indicate that they tested 12 samples that were cross-reactive to H5N1 viruses by MFI to confirm seropositivity. The MFI measures binding antibodies and humans can develop cross-reactive HA head antibodies. Given that H5N1 viruses are not circulating in humans, it more likely that patients in the cohort have cross reactive HA head binding antibodies than they were infected with H5N1. Thus, the authors are actually confirming seronegativity. Please revise this section for accuracy.

Response:

To clarify this point, our study addressed 2 questions:

- 1) What are the levels of population immunity to A(H5N1) viruses in the US population?
To answer this question, we conducted comprehensive assessments of the levels of neutralizing antibodies, HA head binding antibodies, functional NA inhibition antibodies (NAI), NA binding antibodies to HPAI clade 2.3.4.4b A(H5N1) viruses, and antibodies to group 1 HA stalk that could contribute to the protection immunity against A(H5N1) viruses in the US population.*
- 2) Are there serological evidence of infection with A(H5N1) among the population surveyed?
Based on WHO A(H5) case definition, if a serum sample contains neutralizing antibodies plus hemagglutination inhibition antibodies or HA head binding*

antibodies “specific” to A(H5N1) viruses, then it is “seropositive” to A(H5), indicating the person has been infected with A(H5N1) viruses: because A(H5N1) specific antibodies can only be induced by infection and not from cross-reactivity with seasonal Influenza viruses. In this paragraph, we are presenting our results based on the WHO’s case definition of “seropositivity” A(H5N1) (The WHO case definition does not define “Seronegativity”). Our results showed there are no seropositivity with no serologic evidence of A(H5) infection among all participants surveyed.

To improve accuracy and clarity, we have revised this section as follows (Line 106-119 in track changed version):

“ Based on the World Health Organization (WHO) A(H5) case definition¹⁷, seropositivity to A(H5) virus using a single serum sample was defined as a neutralizing antibody titer ≥ 40 to an A(H5) virus that is antigenically similar to the viruses that are causing the current outbreak, and either hemagglutination inhibition (HI) titer to an antigenically similar A(H5) virus ≥ 40 , or a positive result from an A(H5)-specific immunological assay such as a multiplex antibody binding assay. Seropositive sera that have antibodies specific to A(H5N1) viruses indicate serological evidence of infection. To confirm A(H5N1) seropositivity, 12 samples from the 2021-23 population immunity study (Fig 1) that had the highest MFI values to 2.3.4.4b A(H5N1) AW/2021 GH were tested by both MN and HI assays against wild type 2.3.4.4b HPAI A(H5N1) AW/2021 virus and 2.3.4.4b HPAI A(H5N1) A/Texas/37/2024 virus. All sera were negative (<10) against both wild type A(H5N1) viruses in both assays, although most had pre-existing neutralizing and HI antibodies to circulating seasonal A(H1N1)pdm09 viruses (Table 2). These results confirmed no A(H5N1) seropositivity among 1467 US population immunity sera surveyed across 3 time points during 2021-23 seasons (Fig 1), with no serological evidence of A(H5N1) infection. “

2) Line 148 indicates “We analyzed the prevalence of antibodies towards the N1 neuraminidase of two 2.3.4.4b viruses”. It would be more accurate to state “cross-reactive antibodies”.

Response:

We revised this sentence as suggested, it now reads (Line 153 in track changed version):

“We analyzed the prevalence of cross-reactive antibodies to the N1 neuraminidases of two 2.3.4.4b viruses in sera collected from 2023-24 participants”

- 3) Line 184 indicates “to elucidate the source of the anti HA stalk antibodies”. The approach used does not “elucidate” the source. Please revise.

Response:

We revised this sentence (Line 189 in track changed version):

“Lastly, to investigate the source of the anti HA stalk antibodies, we analyzed antibody responses to group 1 HA stalk following seasonal influenza vaccination and infection”

- 4) Line 197-198 indicates the “study provides a timely and comprehensive assessment of population immunity in the US to 2.3.4.4b A(H5N1) viruses.” This is not accurate. Instead, the study provides an assessment of cross-reactive immunity present in humans.

Response:

We revised this sentence as suggested: (Line 203-204 in track changed version)

“Our study provides a timely and comprehensive assessment of the cross-reactive immunity present in the US population to 2.3.4.4b A(H5N1) viruses. “

- 5) Line 220, please revise for accuracy. This sentence indicates vaccination, or infection confers prophylactic protection against H5N1 infection in mice, ferrets, and humans. This is not known for humans. Please revise.

Response:

Thanks for the careful review, we have revised this sentence as suggested (Line 225-227 in track changed version):

“Past studies have demonstrated that seasonal influenza virus infection and vaccination could induce cross-reactive and protective NA-specific antibodies, conferring prophylactic protection against older avian A(H5N1) viruses in mice and ferret models”

- 6) Line 233, please provide references indicating NA immunity is protective.

Response:

We added 3 references to support the protective immunity of NA antibodies:

Ref 20: Monto, A. S., et al. (2015). "Antibody to Influenza Virus Neuraminidase: An Independent Correlate of Protection." J Infect Dis.

Ref 29: Koutsakos, M., et al. (2025). "Binding antibody titers against the hemagglutinin and neuraminidase correlate with protection against medically attended influenza A and B disease." *J Virol* 99(6): e0039125.

Ref 30: Eichelberger, M. C., et al. (2018). "Neuraminidase as an influenza vaccine antigen: a low hanging fruit, ready for picking to improve vaccine effectiveness." *Curr Opin Immunol* 53: 38-44.

Ref 20, 29, 30 (Line 240 in track changed version), the sentence now reads::

"Nevertheless, the role of neuraminidase antibodies in influenza protective immunity has been well recognized(Monto, Petrie et al. 2015, Eichelberger, Morens et al. 2018, Koutsakos, Reynaldi et al. 2025)."

- 7) Line 249, several ferret studies have been published evaluating cross protective immunity to seasonal viruses against H5N1. Please include these references.

Response:

Good point. Since the bovine A(H5N1) outbreak, there are several recent ferret studies published that evaluated the impact of pre-existing immunity to seasonal Influenza on the cross-protection against 2.3.4.4b A(H5N1) challenge. We have already included reference 13, and have added additional references ref 14 and ref 40.

Ref 13: Sun, X., et al. (2025). "Effect of Prior Influenza A(H1N1)pdm09 Virus Infection on Pathogenesis and Transmission of Human Influenza A(H5N1) Clade 2.3.4.4b Virus in Ferret Model." *Emerg Infect Dis* 31(3): 458-466.

Ref 14: Le Sage, V., et al. (2025). "Influenza A(H5N1) Immune Response among Ferrets with Influenza A(H1N1)pdm09 Immunity." *Emerg Infect Dis* 31(3): 477-487.

Ref 30: Restori, K. H., et al. (2025). "Preexisting immunity to the 2009 pandemic H1N1 virus reduces susceptibility to H5N1 infection and disease in ferrets." *Sci Transl Med* 17(808): eadw4856.

We also added an additional sentence in the discussion (Line 256-257 in track changed version):

"Several recent studies demonstrated the impact of pre-existing immunity from seasonal influenza viruses on the cross-protection against A(H5N1) challenge in the ferret model^{13, 14, 30"}

- 8) Lines 258-268. Please revise this section to clarify that vaccination prevents influenza illness and disease, and not infection.

Response:

We have revised the sentence below as suggested (Line 275-278 in track changed version):

“While seasonal influenza vaccines are not intended to prevent A(H5) influenza infection, our data from humans revealed that they could induce cross-reactive and potentially cross-protective NA and HA stalk antibodies; the clinical implications of these responses in A(H5) human infection merit further study. “

- 9) Table 2. It would strengthen the dataset to add neutralizing titers in the human serum against a circulating H1N1 virus.

Response:

Excellent suggestion.

In the revised table 2, we have added not only the neutralizing titers, but also the MFI and HI antibody titers to representative circulating A(H1N1)pdm09 viruses during the study season as a seasonal influenza virus control.